# Macrophage inflammation resolution requires CPEB4-directed offsetting of mRNA degradation

Clara Suñer[1†], Annarita Sibilio[1†], Judit Martín[1], Chiara Lara Castellazzi[1], Oscar Reina[1], Ivan Dotu[2‡], Adrià Caballé[1], Elisa Rivas[1], Vittorio Calderone[1], Juana Díez[2], Angel R Nebreda[1,3]*, Raúl Méndez[1,3]*

[1]Institute for Research in Biomedicine (IRB Barcelona), The Barcelona Institute of Science and Technology, Barcelona, Spain; [2]Universitat Pompeu Fabra, Barcelona, Spain; [3]Institució Catalana de Recerca i Estudis Avançats (ICREA), Barcelona, Spain

**\*For correspondence:**
angel.nebreda@irbbarcelona.org (ARN);
raul.mendez@irbbarcelona.org (RM)

[†]These authors contributed equally to this work

**Present address:** [‡]Moirai Biodesign, Barcelona, Spain

**Competing interest:** The authors declare that no competing interests exist.

**Abstract** Chronic inflammation is a major cause of disease. Inflammation resolution is in part directed by the differential stability of mRNAs encoding pro-inflammatory and anti-inflammatory factors. In particular, tristetraprolin (TTP)-directed mRNA deadenylation destabilizes AU-rich element (ARE)-containing mRNAs. However, this mechanism alone cannot explain the variety of mRNA expression kinetics that are required to uncouple degradation of pro-inflammatory mRNAs from the sustained expression of anti-inflammatory mRNAs. Here, we show that the RNA-binding protein CPEB4 acts in an opposing manner to TTP in macrophages: it helps to stabilize anti-inflammatory transcripts harboring cytoplasmic polyadenylation elements (CPEs) and AREs in their 3'-UTRs, and it is required for the resolution of the lipopolysaccharide (LPS)-triggered inflammatory response. Coordination of CPEB4 and TTP activities is sequentially regulated through MAPK signaling. Accordingly, CPEB4 depletion in macrophages impairs inflammation resolution in an LPS-induced sepsis model. We propose that the counterbalancing actions of CPEB4 and TTP, as well as the distribution of CPEs and AREs in their target mRNAs, define transcript-specific decay patterns required for inflammation resolution. Thus, these two opposing mechanisms provide a fine-tuning control of inflammatory transcript destabilization while maintaining the expression of the negative feedback loops required for efficient inflammation resolution; disruption of this balance can lead to disease.

## Editor's evaluation

This article by Suner et al. investigated specific regulation of LPS-induced sepsis and mechanisms underlying resolution of inflammation. The authors focused on CPEB4 and TTP, two RNA-binding proteins, showing that they work in opposite manner to regulate RNA stability in macrophages and modulate inflammation. This is an interesting study that offers significant advance to the fields of sepsis-inflammation and post-transcriptional regulation of gene expression in inflammatory responses.

## Introduction

As part of the innate immune system, macrophages sense infectious pathogens and orchestrate inflammatory and antimicrobial immune responses. These immune reactions require tight temporal regulation of multiple pro- and anti-inflammatory factors, connected by negative feedback loops, which ultimately ensure inflammation resolution. Thus, these gene expression programs are dynamic and coordinately regulated, and alterations of these can cause pathological inflammation and disease,

including autoimmune disorders and cancer (*Carpenter et al., 2014*). While changes in the rate of gene transcription are important for an initial inflammatory response, the duration and strength of the response are determined mainly by the rate of mRNA decay (*Rabani et al., 2011*).

AU-rich elements (AREs) are key *cis*-acting elements that regulate mRNA deadenylation and the stability of transcripts involved in inflammation (*Spasic et al., 2012*). The role of the ARE-binding protein tristetraprolin (TTP) has been widely characterized in macrophages that have been stimulated by bacterial lipopolysaccharide (LPS) (*Carpenter et al., 2014*), which is the main membrane component of Gram-negative bacteria. During late response to LPS, TTP-mediated mRNA decay limits the expression of inflammatory genes, thereby establishing a post-transcriptional negative feedback loop that promotes inflammation resolution (*Anderson, 2010*; *Spasic et al., 2012*). However, during early response to LPS, TTP activity is counterbalanced by another ARE-binding protein, Hu-antigen R (HuR), which stabilizes its mRNA targets by competing with TTP for ARE occupancy (*Tiedje et al., 2012*). The competitive binding equilibrium between TTP and HuR is post-translationally regulated by the mitogen-activated protein kinase (MAPK) signaling pathways, whose activation is induced by LPS through Toll-like receptor 4 (TLR4) (*Arthur and Ley, 2013*; *O'Neil et al., 2018*).

Of note, ARE-mediated deadenylation and mRNA decay alone cannot explain the variety of temporal expression patterns and destabilization kinetics needed for inflammation resolution (*Sedl-yarov et al., 2016*). Thus, additional mechanisms are likely required to define the temporal expression patterns of pro- and anti-inflammatory genes. In early development, the cytoplasmic regulation of poly(A) tail length is not unidirectional but rather a dynamic equilibrium between deadenylation and polyadenylation by ARE-binding proteins and cytoplasmic polyadenylation element binding proteins (CPEBs) (*Belloc and Méndez, 2008*; *Nousch et al., 2019*; *Piqué et al., 2008*). CPEBs bind the cytoplasmic polyadenylation elements (CPEs) present in the 3′ untranslated region (UTR) of some mRNAs and promote the translation of these mRNAs by favoring elongation of their poly(A) tails (*Ivshina et al., 2014*; *Weill et al., 2012*). The activities exerted by CPEBs are quantitatively defined by the number and position of CPEs present in their target transcripts, thus determining the polyadenylation kinetics and the transcript-specific temporal patterns of translation.

The global contribution of CPEBs to the regulation of mRNA expression during inflammatory processes has not been addressed. In this work, we show that myeloid CPEB4 is needed for the resolution of the LPS-triggered inflammatory response. We further show that the levels and activities of CPEB4 and TTP are sequentially regulated by LPS-induced MAPK signaling. In turn, the combination of CPEs and AREs in their target mRNAs generates transcript-specific decay rates. We propose that these two opposing mechanisms allow destabilization of inflammatory transcripts while maintaining the expression of the negative feedback loops required for efficient inflammation resolution.

## Results

### Inflammation resolution is impaired in *Cpeb4*⁻/⁻ macrophages

To determine a potential contribution of CPEBs to the regulation of inflammatory responses, we interrogated the expression of *Cpeb*-encoding mRNAs during the course of a systemic inflammatory response in sepsis patients. In two independent GEO datasets, we observed that *Cpeb4* mRNA levels were significantly upregulated in blood, whereas the *Cpeb1–3* mRNAs did not present major expression changes (*Figure 1A*). The peripheral blood of sepsis patients displays an increased number of monocytes/macrophages, which are one of the immune cell populations that express higher *Cpeb4* mRNA levels (*Figure 1—figure supplement 1*). Moreover, deconvolution analysis suggested that myeloid cells expressed higher levels of *Cpeb4* mRNA during sepsis (*Figure 1—figure supplement 1*). To further explore the significance of this correlation, we generated a Cre-mediated knockout mouse line of *Cpeb4* specific for myeloid cells (*Cpeb4ˡᵒˣ/ˡᵒˣLyz2Cre*, hereafter referred to as Cpeb4MKO); of note, these mice did not show any major phenotype in homeostatic conditions (*Figure 1—figure supplement 2*). Upon challenging these mice with an intraperitoneal dose of LPS (for LPS-induced endotoxic shock), Cpeb4MKO mice displayed lower survival rates than the wildtype (WT) controls (*Figure 1B*). Cpeb4MKO animals also presented splenomegaly (*Figure 1C*) and increased splenic levels of the cytokines *Il6*, *Tnf*, and *Il1a* (*Figure 1D*), key in septic response (*Schulte et al., 2013*). These results link CPEB4 ablation in myeloid cells to an exacerbated inflammatory response, which impaired their survival when faced with sepsis.

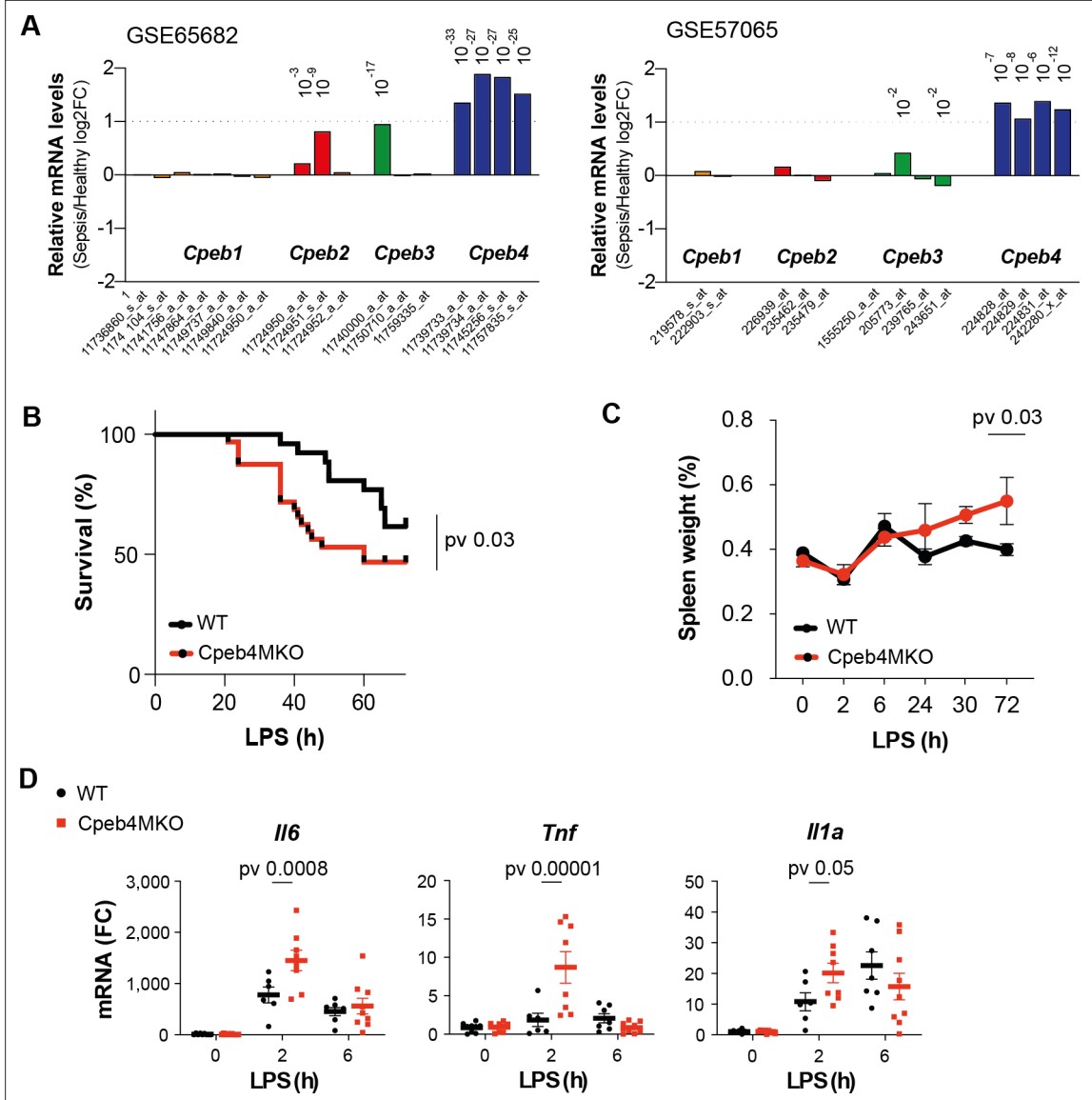

**Figure 1.** CPEB4 downregulation in myeloid cells increases sepsis-induced mortality. (**A**) Differential expression of *Cpeb* mRNAs in the blood of sepsis patients/healthy individuals. Statistics: limma-moderated *t*-test. Pvadj (Benjamini–Hochberg) is shown. (**B–D**) Wildtype (WT) and myeloid-specific *Cpeb4* KO mice (Cpeb4MKO) were injected with an i.p. dose of lipopolysaccharide (LPS). (**B**) Kaplan–Meier survival curves. Results represent three independent experiments (n > 7/group/experiment). Statistics: likelihood-ratio test p-value. (**C**) Spleen weights normalized to total animal weight. Statistics: two-way ANOVA. (**D**) Splenic total mRNA was measured by RT-qPCR and referred to *Tbp* (n > 5 animals/condition). Statistics: multiple *t*-test. (**C, D**) Data are represented as mean ± SD.

The online version of this article includes the following figure supplement(s) for figure 1:

**Figure supplement 1.** CPEB4 upregulation in myeloid cells in septic patients.

**Figure supplement 2.** Characterization of Cpeb4MKO mice.

To further characterize CPEB4 function during the LPS response, we stimulated bone marrow-derived macrophages (BMDMs) with LPS. Time-course analysis showed that *Cpeb4* mRNA (but no other *Cpeb* mRNA) was transiently upregulated, peaking between 3 and 6 hr after LPS stimulation (***Figure 2A***), followed by CPEB4 protein accumulation (***Figure 2B***, ***Figure 2—figure supplement 1***). The CPEB4 protein was detected as a doublet, with the slow migrating band corresponding to the hyperphosphorylated, active form (***Guillén-Boixet et al., 2016***; ***Figure 2C***). Isolated BMDMs from *Cpeb4* total KO mice (hereafter, *Cpeb4$^{-/-}$*) had no major defects in their differentiation or proliferation rates compared with WT BMDMs, based on marker genes (***Figure 2—figure supplement 2***). In

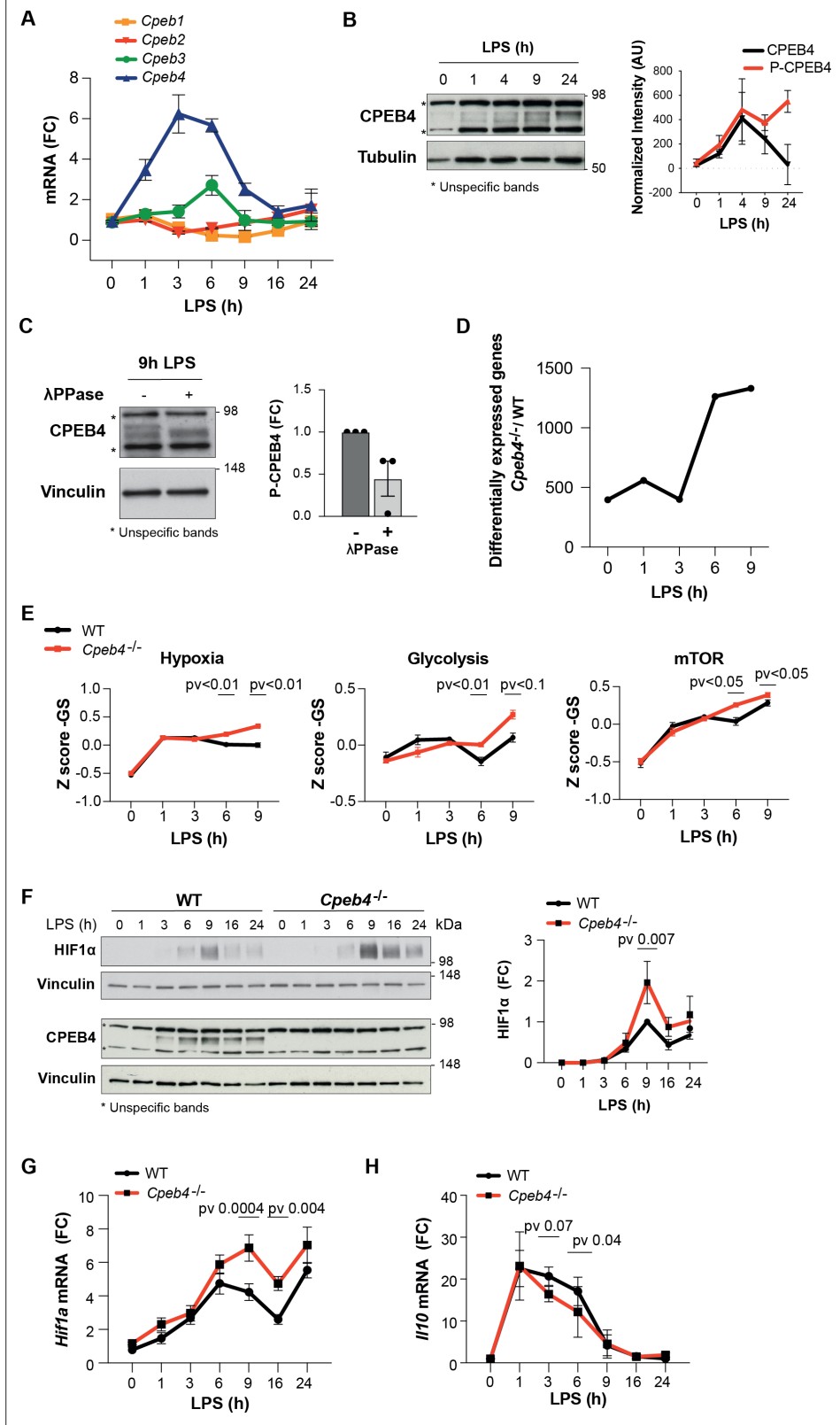

**Figure 2.** Inflammation resolution is impaired in *Cpeb4⁻/⁻* macrophages. (**A–C**) Lipopolysaccharide (LPS)-stimulated wildtype (WT) bone marrow-derived macrophages (BMDMs). (**A**) *Cpeb1–4* levels were measured by RT-qPCR (n = 6). (**B,** left) CPEB4 immunoblot, using α-tubulin as loading control. (Right) CPEB4 quantification normalized to α-tubulin (n = 3; data shown in ***Figure 2—figure supplement 2***). (**C**, left) CPEB4 immunoblot in protein extracts

*Figure 2 continued on next page*

*Figure 2 continued*

treated with λ phosphatase when indicated. (Right) Quantification of P-CPEB4 signal (n = 3). (**D–H**) LPS-stimulated WT and *Cpeb4*−/− BMDMs. (**D**) Number of differentially expressed genes (p<0.01) between genotypes. mRNA levels were quantified by RNAseq (n = 4). Statistics: DESeq2 R package. (**E**) Z-score signature of the indicated pathways. mRNA levels were quantified by RNAseq (n = 4). Statistics: rotation gene set enrichment analysis. (**F**, left) HIF1a and CPEB4 immunoblot, vinculin served as loading control. (Right) Normalized quantification, signal intensity was normalized to vinculin and fold change to WT at 9 hr after LPS induction was calculated (n = 3; data shown in *Figure 2—figure supplement 2*). (**G, H**) *Hif1a* and *Il10* levels measured by RT-qPCR (n = 6). (**A, G**) *Tbp* was used to normalize. (**B, C, E, F**) Data are represented as mean ± SEM. (**F–H**) Statistics: two-way ANOVA. (**D, E**) See also *Supplementary file 1*.

The online version of this article includes the following source data and figure supplement(s) for figure 2:

**Source data 1.** Blots corresponding to *Figure 2B* and *Figure 2—figure supplement 1A*.

**Source data 2.** Blots corresponding to *Figure 2C*.

**Source data 3.** Blots corresponding to *Figure 2F* and *Figure 2—figure supplement 3A*.

**Figure supplement 1.** CPEB4 upregulation in lipopolysaccharide (LPS)-stimulated macrophages.

**Figure supplement 2.** Differentiation of *Cpeb4*−/− bone marrow derived macrophages.

**Figure supplement 3.** Characterization of lipopolysaccharide (LPS) response in *Cpeb4*−/− macrophages.

---

contrast, and consistent with the observation that phosphorylated CPEB4 accumulated during the late phase of the LPS response, comparative transcriptomic analysis between WT and *Cpeb4*−/− BMDMs showed that most differences appeared 6–9 hr after LPS stimulation, during the resolution phase of the inflammatory response (*Figure 2D*, *Supplementary file 1*). The transcriptomic profile of *Cpeb4*−/− BMDMs included increases in hypoxia, glycolysis, and mTOR pathways, all of which have been linked to pro-inflammatory macrophage polarization during sepsis (*McGettrick and O'Neill, 2020*; *Shalova et al., 2015*; *Figure 2E*). Further, we confirmed that HIF1a levels were increased, both at protein and mRNA levels, after 9 hr of LPS treatment (*Figure 2F and G*, *Figure 2—figure supplement 3*). Interferon, NOTCH or IL6 signaling pathways were affected in the KO (*Figure 2—figure supplement 3*). Conversely, the levels of anti-inflammatory transcripts, such as *Il10* mRNA, were reduced in the *Cpeb4*−/− BMDMs (*Figure 2H*). Altogether, these results suggest that the absence of CPEB4 disrupts the development and resolution of the LPS-induced inflammatory response in myeloid cells.

## The p38α-HuR-TTP axis regulates *Cpeb4* mRNA stability

We next addressed how CPEB4 expression is regulated in BMDMs treated with LPS. Since the peak of *Cpeb4* mRNA followed similar kinetics to pro-inflammatory ARE-containing mRNAs, such as *Il6* and *Il1a* (*Figure 3—figure supplement 1*), we searched the 3′-UTR of *Cpeb4* for AREs; we found 17 repeats of the AUUUA pentanucleotide (*Figure 3—figure supplement 1*). ARE-containing transcripts are stabilized through HuR binding during the early phase of the LPS response and destabilized at later times by TTP binding. The switch between these two ARE-binding proteins is regulated by the LPS-activated p38α MAPK, which is encoded by *Mapk14* (*Tiedje et al., 2012*). Next, we analyzed BMDMs from myeloid-specific *Mapk14* KO mice (*Mapk14*lox/lox*Lyz2*Cre, hereafter referred to as p38αMKO) (*Youssif et al., 2018*) and found that *Cpeb4* mRNA expression levels were indeed reduced in LPS-treated p38αMKO macrophages (*Figure 3A*), suggesting that *Cpeb4* expression is controlled by the p38α-HuR-TTP axis. This observation was confirmed by treating WT BMDMs with the p38α inhibitor PH797804 (*Figure 3B*). To test whether p38α regulates *Cpeb4* expression at the transcriptional or post-transcriptional level, we first inhibited transcription in LPS-treated WT and p38αMKO BMDMs; we found that *Cpeb4* mRNA levels decayed significantly faster in the latter (*Figure 3C*, *Figure 3—figure supplement 2*). We then assessed whether the stabilization of *Cpeb4* mRNA by p38α was mediated through the differential binding of HuR and TTP. We observed that HuR binding to *Cpeb4* mRNA was strongly enriched after LPS treatment in WT BMDMs but to a lesser extent in p38αMKO BMDMs, as seen in RNA-immunoprecipitation experiments (*Figure 3D*, *Figure 3—figure supplement 3*), indicating that this induced binding occurred in a p38α-dependent manner. Similar results were observed in *Tnf* mRNA as a control (*Figure 3D*). We then analyzed recently published datasets from TTP immunoprecipitation (iCLIP) followed by genome-wide analysis of its associated mRNAs in LPS-stimulated BMDMs (*Sedlyarov et al., 2016*). We observed binding of TTP to the *Cpeb4* 3′-UTR

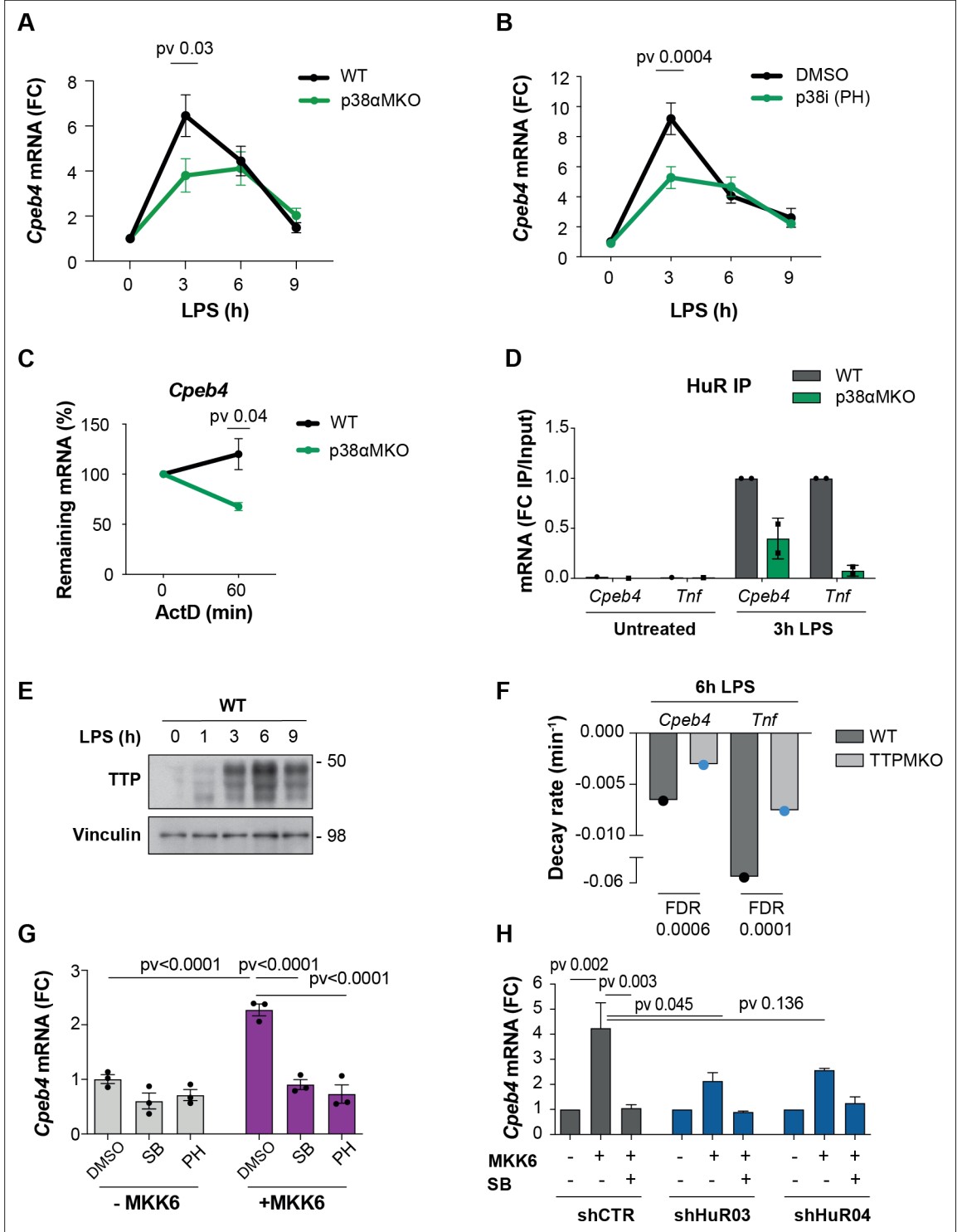

**Figure 3.** The p38α-HuR-TTP axis regulates *Cpeb4* mRNA stability. (**A**) *Cpeb4* levels in wildtype (WT) and p38αMKO bone marrow-derived macrophages (BMDMs) stimulated with lipopolysaccharide (LPS) (n = 3). (**B**) *Cpeb4* levels in LPS-stimulated BMDMs treated with the p38α inhibitor PH-797804 (or DMSO as control) (n = 4). (**C**) WT or p38αMKO BMDMs were stimulated with LPS for 1 hr; *Cpeb4* mRNA stability was measured after treating with actinomycin D (ActD). Statistics: paired *t*-test (60 min time point; n = 3). See also *Figure 3—figure supplement 2*. (**D**) *Cpeb4* mRNA levels in HuR RNA-immunoprecipitates (IP) performed in WT or p38αMKO BMDMs, after LPS stimulation as indicated. IgG IPs served as control. IP/input enrichment is shown, normalized to WT IP LPS (n = 2). See also *Figure 3—figure supplement 3*. (**E**) Immunoblot of TTP protein in WT BMDMs treated with LPS for 0–9 hr. Vinculin served as loading control (n = 2). (**F**) *Cpeb4* and *Tnf* decay rates in WT and TTPMKO BMDMs stimulated for 6 hr with LPS (data from *Sedlyarov et al., 2016*). Data represents the mean of three biological replicates. (**G, H**) U2OS cells were treated with tetracycline to induce

*Figure 3 continued on next page*

*Figure 3 continued*

the expression of a constitutively active MKK6, which induces p38α MAPK activation (***Trempolec et al., 2017***). (**G**) *Cpeb4* levels upon p38α activation (+MKK6) or inhibition with SB203580 or PH-797804 (n = 3). (**H**) *Cpeb4* levels in control or HuR-depleted U2OS cells, where p38α MAPK signaling has been activated (+MKK6) or inhibited (SB) (n = 2). See also ***Figure 3—figure supplement 5***. (**A–D, G, H**) mRNA levels were quantified by RT-qPCR. *Gapdh* (**A, B, C, G**) was used to normalize. (**A, B, D, G, H**) Data are represented as mean ± SEM. (**A, B**) Statistics: two-way ANOVA. (**C**) Statistics: paired *t*-test. (**G, H**) Statistics: one-way ANOVA, selected pvadj are shown.

The online version of this article includes the following source data and figure supplement(s) for figure 3:

**Source data 1.** Blots corresponding to *Figure 3E*.

**Figure supplement 1.** *Cpeb4* mRNA as a target of the p38α-HuR-TTP axis.

**Figure supplement 2.** mRNA stability in wildtype (WT) and p38αMKO bone marrow-derived macrophages (BMDMs).

**Figure supplement 3.** HuR Immunoprecipitation (IP) in wildtype (WT) and p38αMKO bone marrow-derived macrophages (BMDMs).

**Figure supplement 4.** Tristetraprolin (TTP) binds *Cpeb4* mRNA.

**Figure supplement 5.** The p38α-HuR axis regulates *Cpeb4* mRNA.

at 6 hr after LPS treatment (***Figure 3—figure supplement 4***) when the TTP protein levels were high (***Figure 3E***), as well as reduced decay rates of both *Cpeb4* and *Tnf* mRNAs in macrophages from mice with a myeloid cell-specific TTP deletion (hereafter, TTPMKO) (***Figure 3F***). To further analyze the contribution of p38α signaling to *Cpeb4* mRNA stability, we used human osteosarcoma (U2OS) cells expressing the p38α MAPK activator MKK6 under the control of the TET-ON promoter (***Trempolec et al., 2017***). MKK6-induced p38α activation was sufficient to increase *Cpeb4* mRNA levels in these cells, and this effect was reversed by treatment with p38α chemical inhibitors (***Figure 3G***) as well as by shRNA-mediated depletion of HuR (***Figure 3H***, ***Figure 3—figure supplement 5***). Altogether, these results indicate that the *Cpeb4* mRNA was stabilized by HuR and destabilized by TTP, and that the demonstrated competitive binding (***Mukherjee et al., 2014***; ***Tiedje et al., 2012***) of these two ARE-binding proteins to *Cpeb4* mRNA is regulated by LPS-driven p38α activation.

## CPEB4 stabilizes mRNAs that encode for negative feedback regulators of the LPS response

We next addressed the functional contribution of CPEB4 to inflammation resolution by performing RIP-seq to identify CPEB4-bound mRNAs in untreated or LPS-activated WT BMDMs, using *Cpeb4*⁻/⁻ BMDMs as controls (***Figure 4A***). We identified 1173 and 1829 CPEB4-associated mRNAs in untreated and LPS-treated BMDMs, respectively (***Supplementary file 2***). These included previously described CPEB4 targets such as *Txnip* or *Vegfa*, while negative controls such as *Gapdh* were not detected (***Figure 4B***). The 3′-UTRs of the CPEB4 co-immunoprecipitated mRNAs were enriched in canonical CPE motifs (***Piqué et al., 2008***), which we previously described as CPEB4 binding elements (***Afroz et al., 2014***; ***Igea and Méndez, 2010***; ***Novoa et al., 2010***), as well as in CPEs containing A/G substitutions (***Figure 4C***, ***Figure 4—figure supplement 1***). These CPE variants have been shown to be specifically recognized by the *Drosophila* ortholog of CPEB2-4 (Orb2) (***Stepien et al., 2016***). Gene Ontology analysis of CPEB4 targets indicated an enrichment of mRNAs encoding for components of the LPS-induced MAPK pathways (***Figure 4—figure supplement 1***). These targets included mRNAs that participate in anti-inflammatory feedback loops that negatively regulate the LPS response, consistent with the phenotype observed in *Cpeb4*⁻/⁻ BMDMs. Thus, *Dusp1, Il1rn, Socs1, Socs3, Zfp36* (which encodes TTP), and *Tnfaip3* mRNAs specifically co-immunoprecipitated with CPEB4 (***Figure 4D***). CPEB4 binding to *Txnip, Dusp1,* and *Il1rn* mRNAs was further validated by RT-qPCR, using *Gapdh* as negative control (***Figure 4E***). The functional importance of the CPEB4 association was confirmed by the observation that both the *Socs1* mRNA (***Figure 4F***) and SOCS1 protein (***Figure 4G***) levels were reduced in LPS-treated *Cpeb4*⁻/⁻ BMDMs compared to WT BMDMs. Likewise, the levels of the *Dusp1, Il1rn, Socs3, Tnfaip3,* and *Zfp36* mRNAs were reduced in LPS-treated *Cpeb4*⁻/⁻ BMDMs (***Figure 4H***, ***Figure 4—figure supplement 2***). To examine whether these changes in mRNA levels originated transcriptionally or post-transcriptionally, we measured the stability of *Socs1* and *Il1rn* mRNAs in WT or *Cpeb4*⁻/⁻ BMDMs stimulated with LPS and treated with actinomycin D to block transcription. Indeed, both *Socs1* and *Il1rn* mRNAs displayed reduced stability in the absence of CPEB4 (***Figure 4I***), showing a strong element of post-transcriptional regulation of the mRNA levels. To confirm that this effect was mediated by the presence of CPEs in the mRNA 3′-UTRs, we studied the stability of reporter mRNAs

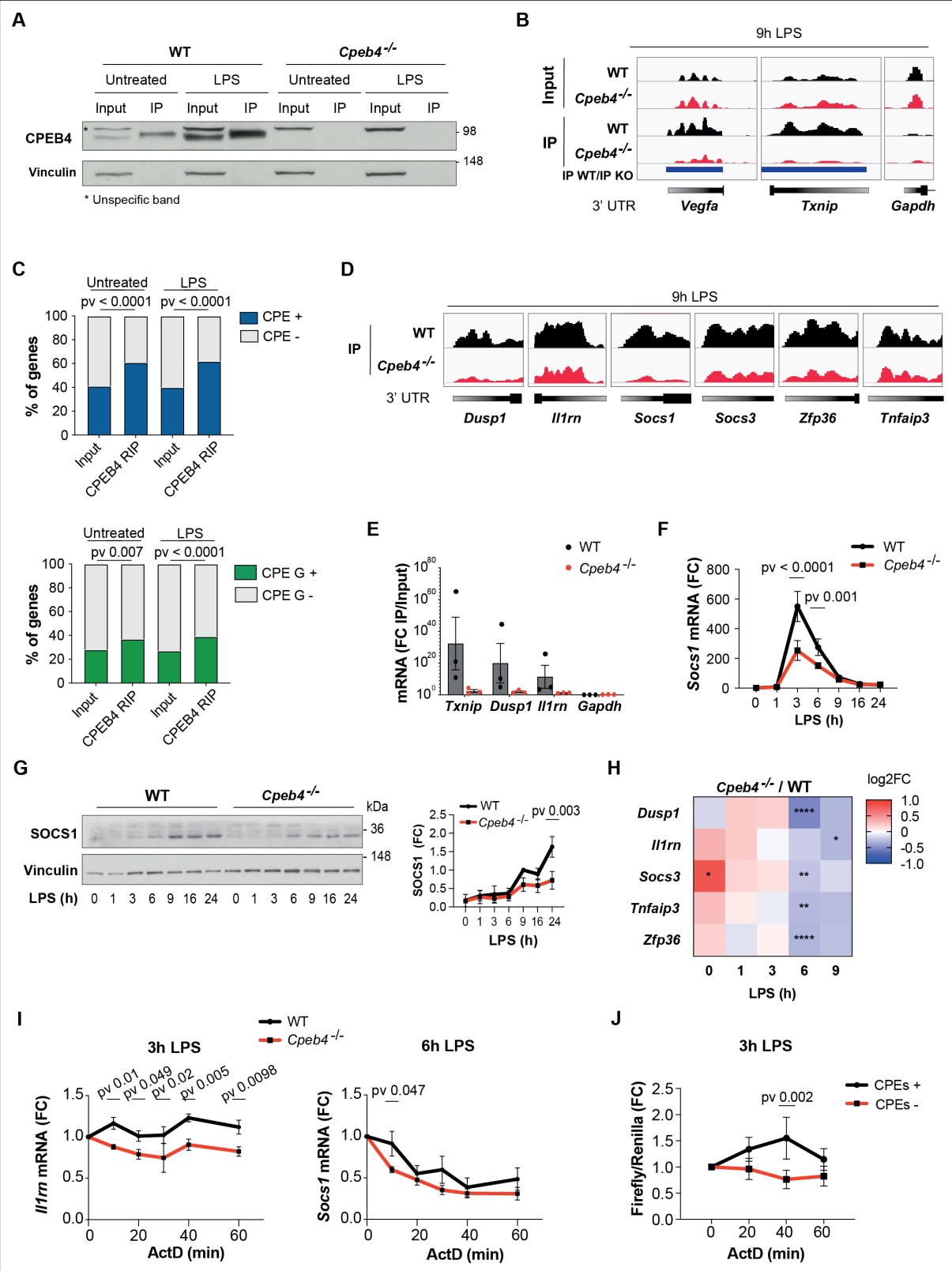

**Figure 4.** CPEB4 stabilizes mRNAs encoding negative feedback regulators of the lipopolysaccharide (LPS) response. (**A–D**) CPEB4 RNA-Immunoprecipitation (IP) and sequencing was performed using total lysates (input) from wildtype (WT) or *Cpeb4⁻/⁻* bone marrow-derived macrophages (BMDMs) that had been treated or not with LPS for 9 hr (n = 1). (**A**) CPEB4 immunoblot, using vinculin as a loading control. (**B**) Examples of read coverage of input or IP of selected mRNAs. Peak enrichments between WT and *Cpeb4⁻/⁻* IPs are shown in blue. (**C**) Cytoplasmic polyadenylation

*Figure 4 continued on next page*

*Figure 4 continued*

element (CPE) and CPE G-containing transcripts according to *Piqué et al., 2008*) in input and CPEB4 IPs. The script from *Piqué et al., 2008* was modified to consider TTTTGT as a CPE motif. Statistics: Fisher's exact test. (**D**) Read coverage of IPs of selected mRNAs. (**E**) CPEB4 IP and RT-qPCR were performed for WT or *Cpeb4⁻/⁻* BMDMs stimulated with LPS for 9 hr. IP/input enrichment is shown (n = 3). (**F**) *Socs1* mRNA levels in LPS-stimulated WT and *Cpeb4⁻/⁻* BMDMs. mRNA levels were measured by RT-qPCR normalizing to *Tbp* (n = 6). (**G**) Immunoblot of SOCS1 in WT and *Cpeb4⁻/⁻* BMDMs treated with LPS. Vinculin served as loading control. Quantification is shown (FC to WT, after 9 hr of LPS) (n = 3). (**H**) Differential expression between WT and *Cpeb4⁻/⁻* BMDMs treated with LPS measured by RNAseq (n = 4). Statistics: DESeq2 R package. (**I**) mRNA stability was measured by treating with actinomycin D (ActD) WT and *Cpeb4⁻/⁻* BMDMs stimulated with LPS for the indicated times. Gene expression was analyzed by RT-qPCR, normalized to *Gapdh/Tbp* (n = 4). (**J**) RAW 264.7 macrophages were transfected with a Firefly luciferase reporter under the control of the cyclin B1 3'-UTR, containing either WT (CPE+) or mutated (CPE–) CPE motifs. The same plasmid contained Renilla luciferase reporter as a control. Macrophages were stimulated with LPS for 3 hr, at which point ActD was added. mRNA levels were measured by RT-qPCR. (**B, D**) Integrated Genomic Viewer (IGV) images. (**E–G**) Data are represented as mean ± SEM. (**F, G, I, J**) Statistics: two-way ANOVA. See also *Supplementary files 1-2*.

The online version of this article includes the following source data and figure supplement(s) for figure 4:

**Source data 1.** Blots corresponding to *Figure 4A*.

**Source data 2.** Blots corresponding to *Figure 4G*.

**Figure supplement 1.** Characterization of CPEB4 targets in lipopolysaccharide (LPS)-stimulated macrophages.

**Figure supplement 2.** Expression of negative feedback regulators of the lipopolysaccharide (LPS) response is impared in *Cpeb4⁻/⁻* bone marrow-derived macrophages (BMDMs).

with or without CPEs in LPS-stimulated RAW264 macrophages. Indeed, after 3 hr of LPS treatment, the stability of a reporter with CPEs increased compared to the same mRNAs but with CPEs inactivated by point mutations (*Figure 4J*). Taken together, our results suggested that CPEB4 sustained the expression of anti-inflammatory factors post-transcriptionally at late times following LPS stimulation by binding to the corresponding mRNAs and promoting their stabilization.

## The equilibrium between CPEB4/CPEs and TTP/AREs defines transcript levels during inflammation resolution

Given that CPEB4 and its binding element, CPE, have opposite effects on mRNA stability as TTP and its binding element ARE, we explored what happens when both binding elements are present in the same 3'-UTR. We found that CPEB4-bound mRNAs in BMDMs were enriched in AREs (*Figure 5A*, *Supplementary file 3*). Conversely, 53% (102/193) of the TTP-bound mRNAs in BMDMs (*Sedlyarov et al., 2016*; *Supplementary file 4*) were also co-immunoprecipitated by CPEB4 (*Figure 5B*). Indeed, at a genome-wide level, we observed a linear correlation between the number of CPEs (*Piqué et al., 2008*) and the number of AREs (*Gruber et al., 2011*) present in the same 3'-UTR (*Figure 5C*, *Supplementary file 3*). To define how the coexistence of these two elements impacted mRNA stability, we compared the expression kinetics of mRNAs targeted by both CPEB4 and TTP, containing both CPEs and AREs but at different ratios, and the levels of these transcripts in LPS-activated BMDMs from *Cpeb4⁻/⁻* (*Figure 5D*, *Supplementary file 1*) or TTPMKO mice (*Sedlyarov et al., 2016*; *Figure 5E*). We found a wide range of differential responses to TTP or CPEB4 depletion. For example, the induction of *Socs1* and *Il1rn* mRNA was reduced in *Cpeb4⁻/⁻* macrophages, but the expression of *Cxcl1* or *Ptgs2* mRNA remained unaffected. Conversely, in the absence of TTP, the *Socs1* and *Il1rn* mRNAs did not display major changes, while the *Cxcl1* and *Ptgs2* mRNA levels increased. In a third group, mRNA levels (including *Ccl2* and *Cxcl2*) were affected (in opposite directions) in both the *Cpeb4⁻/⁻* and TTPMKO macrophages. Thus, despite the fact that all were bound by both proteins, some mRNAs were more dependent on CPEB4-mediated stabilization, while others were more sensitive to TTP-mediated destabilization. Interestingly, the 3'-UTRs of *Socs1* and *Il1rn* mRNAs were more enriched in CPEs, those of *Cxcl1* and *Ptgs2* mRNA were enriched in AREs, and those of *Ccl2* and *Cxcl2* mRNA presented opposite ARE/CPE ratios.

Based on these observations, we hypothesized that the relative amount of these two *cis*-acting elements (CPEs or AREs) in the 3'-UTR of an mRNA might modulate the relative contribution of TTP and/or CPEB4 to their stability impact of the RNA-binding protein. To test this model, we classified the mRNAs that were targeted by both TTP and CPEB4 according to the $\log_2$ ratio between the number of AREs and CPEs in their 3'-UTRs (ARE:CPE score; *Figure 5F*, *Figure 5—figure supplement 1*). Thus, mRNAs with more CPEs than AREs were classified CPE-dominant (CPE-d, ARE:CPE score <0), and those with more AREs, as ARE-dominant (ARE-d, ARE:CPE score >0) (*Figure 5G*, *Supplementary file*

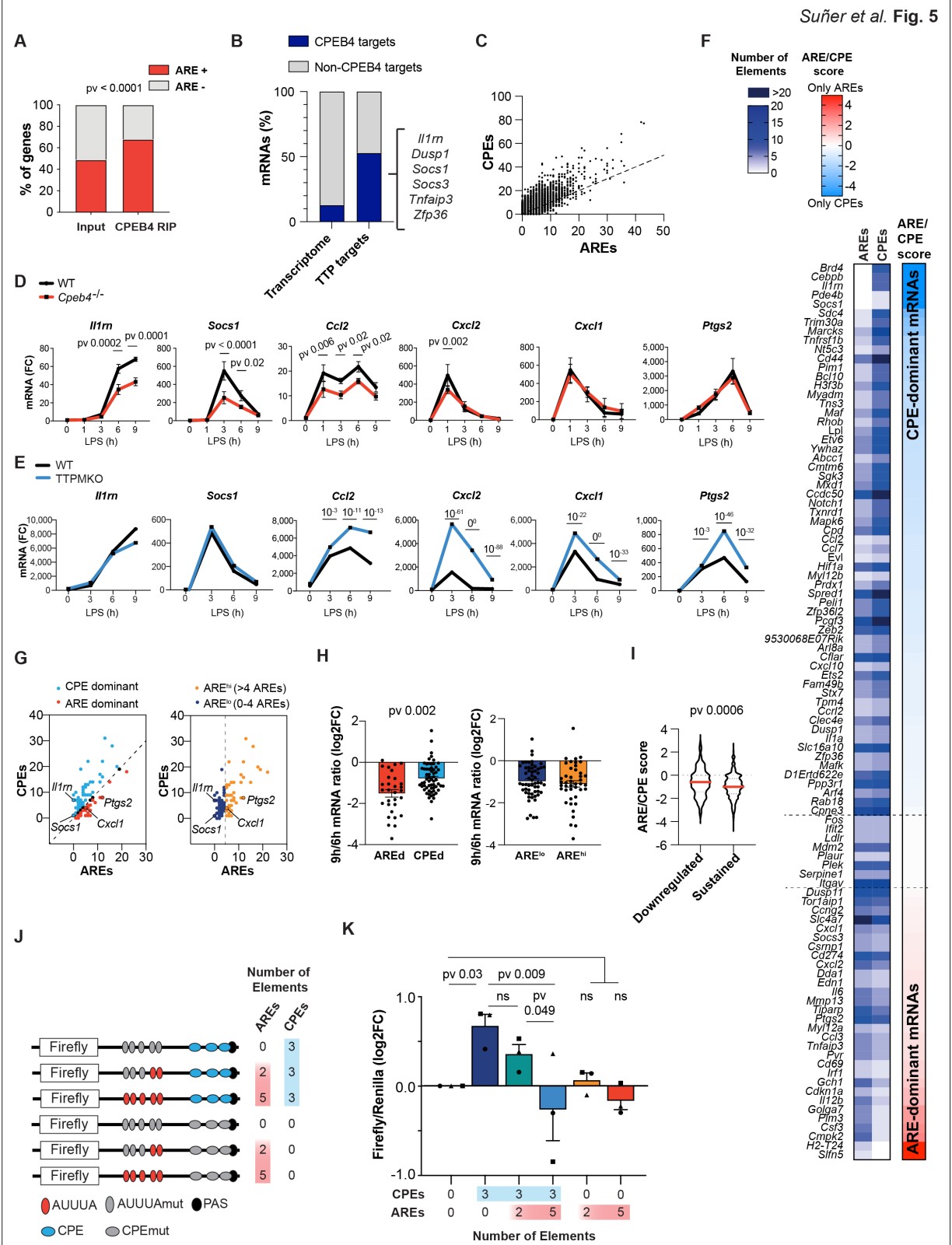

**Figure 5.** The equilibrium between CPEB4/cytoplasmic polyadenylation elements (CPEs) and tristetraprolin (TTP)/AU-rich elements (AREs) defines different transcript-level patterns. (**A**) ARE-containing transcripts in the input and CPEB4 immunoprecipitations (IPs) from *Figure 4A–D*. Statistics: Fisher's exact test. (**B**) Percentage of CPEB4 targets in lipopolysaccharide (LPS)-stimulated bone marrow-derived macrophages (BMDMs) transcriptome and TTP targets in LPS-stimulated BMDMs (*Sedlyarov et al., 2016*). (**C**) Genome-wide correlation between ARE and CPE motifs in 3′-UTRs. The black

*Figure 5 continued*

line shows the linear regression trend line. $R^2$ = 0.6364. (**D**) mRNA levels in wildtype (WT) and *Cpeb4⁻/⁻* BMDMs were measured by RT-qPCR, normalizing to *Tbp* (n = 6). Statistics: two-way ANOVA. *Socs1* data are also shown in *Figure 4F*. (**E**) mRNA expression in WT or TTPMKO BMDMs treated with LPS. Statistics: DESeq2 software, qval is shown (data from *Sedlyarov et al., 2016*). (**F**) Common TTP and CPEB4 target mRNAs were classified according to the ARE:CPE score as ARE-dominant (ARE-d; red; 30 mRNAs) or CPE-dominant (CPE-d; blue; 61 mRNAs) (see also *Figure 5—figure supplement 1*). (**G**) CPEB4 and TTP target mRNAs were plotted according to the number of AREs and CPEs in the 3′-UTR. (Left) The dashed line separates ARE-d and CPE-d mRNAs. (Right) mRNAs were classified according to only the number of AREs in their 3'-UTR. The dashed line separates ARE^high (>4 AREs; yellow; 43 mRNAs) from ARE^low (≤4 AREs; navy; 56 mRNAs) mRNAs. (**H, I**) WT BMDMs were stimulated with LPS and mRNA levels were quantified by RNAseq (n = 4). (**H**) Common CPEB4 and TTP target mRNAs were classified as ARE-d/CPEd (left) or ARE^high/ARE^low (right). For each mRNA, the levels after 9 hr of LPS treatment were normalized by its expression at 6 hr LPS. (**I**) 1521 CPE- and ARE-containing mRNAs were classified as sustained >0.5 (1319 mRNAs) or downregulated <0.5 (202 mRNAs) according to their expression after 9 hr of LPS treatment, after normalizing to the peak of expression throughout the LPS response. For each mRNA, the ARE:CPE score was calculated. (**J, K**) RAW 264.7 macrophages were transfected with a Firefly luciferase reporter under the control of a chimeric 3′-UTR combining Ier3 and cyclin B1 AREs and CPEs motifs, respectively. Inactivating specific CPE or ARE motifs, six different 3′-UTRs with distinct ARE:CPE scores were generated. The same plasmid contained Renilla luciferase reporter as a control. (**J**) Scheme of the six constructs used for the dual luciferase reporter assay. Inactivated motifs are shown in gray. (**K**) RAW 264.7 macrophages were stimulated with LPS for 6 hr and Firefly/Renilla levels were measured by RT-qPCR. Values were normalized to the 0AREs/0CPEs construct. Statistics: one-way ANOVA Friedman test. All significant differences are shown except 5AREs/0CPEs vs. 0AREs/3CPEs (**) and 2AREs/3CPEs (*). (**D, H, K**) Data are represented as mean ± SEM. (**H, I**) Statistics: Mann–Whitney *t*-test. See also *Supplementary files 2-6*.

The online version of this article includes the following figure supplement(s) for figure 5:

**Figure supplement 1.** The equilibrium between CPEB4/cytoplasmic polyadenylation elements (CPEs) and tristetraprolin (TTP)/AU-rich elements (AREs) defines mRNA oscillation patterns.

**Figure supplement 2.** Expression of mRNAs with distinct AU-rich element (ARE)/cytoplasmic polyadenylation element (CPE) score in lipopolysaccharide (LPS)-stimulated macrophages.

*3*). In an expression kinetics analysis following LPS stimulation, we found that the ratio between late time points (6–9 hr), during which transcripts are differentially destabilized, was more pronounced for ARE-d than for CPE-d mRNAs; in contrast, no differences were observed at early time points (1–3 hr) or at late time points if the same mRNAs were classified only as a function of their number of AREs (*Figure 5G and H*, *Figure 5—figure supplement 1*, *Supplementary file 5*). Next, we generated a list of 1521 ARE- and CPE-containing mRNAs (*Supplementary file 6*) and classified them according to their ratio at late time points after LPS stimulation (6–9 hr), as 'downregulated' (6 hr/9 hr ratio <0.5) or 'sustained' (ratio >0.5) mRNAs. Notably, the downregulated mRNAs had a lower ARE:CPE score (*Figure 5I*). However, no differences were found when only the number of AREs were considered (*Figure 5—figure supplement 1*). These results indicate that the ARE:CPE ratio, but not the number of AREs alone, correlated with differential mRNA expression patterns during the resolutive phase of the LPS response.

To directly test this model, we expressed Firefly luciferase reporter mRNAs that contained different combinations of CPEs and AREs in macrophages and measured their levels after exposure to LPS. As a transfection control, we used Renilla luciferase with a control 3′-UTR (without CPEs or AREs). The parental Firefly luciferase reporter included five AREs and three CPEs, which were inactivated by point mutations to generate the following combinations of ARE:CPE numbers: 5:3, 2:3, 0:3, 5:0, 2:0, and 0:0 (null) (*Figure 5J*). Compared with the reporter without CPEs or AREs (null), the presence of CPEs correlated with higher mRNA levels at late times after LPS stimulation, and these levels were progressively reduced by the addition of AREs (increasing ARE:CPE score) (*Figure 5K*). On the other hand, the number of AREs in the absence of CPEs had no significant impact on the reporter expression levels (*Figure 5K*). No significant differences were found at early time points (*Figure 5—figure supplement 2*). These results confirmed that the combination of CPEs and AREs on mRNAs, together with the regulation of their *trans*-acting factors CPEB4 and TTP, modulates mRNA stability in a transcript-specific manner to fine-tune mRNA expression during the late phase of inflammatory processes.

## Discussion

Physiological inflammatory responses rely on both rapid induction and efficient resolution to avoid the development of diseases. Differential regulation of mRNA stability plays a pivotal role in the

uncoupling between the expression of pro- and anti-inflammatory mediators and is key to engaging the negative feedback loops that switch off the inflammatory response (*Schott et al., 2014*).

Our work points to temporal control of inflammation resolution as being regulated by CPEB4-mediated mRNA stabilization, which acts in a coordinated manner with the well-known TTP-driven destabilization of mRNAs. We propose that the opposing activities of positive (CPE) and negative (ARE) *cis*-acting elements in the mRNA 3'-UTRs, together with the regulation of their *trans*-acting factors CPEB4 and TTP, generate temporal stability profiles, which fulfill the functions needed in each phase of the inflammatory process.

Our results indicate that CPEB4 and TTP levels and activities could be integrated during the LPS response through cross-regulation at multiple post-transcriptional and post-translational layers. First, their activities are coordinately regulated by MAPK signaling pathways. Upon LPS stimulation, p38α MAPK signaling regulates TTP phosphorylation, causing a shift in the competitive binding from an equilibrium between HuR and TTP towards a HuR preference, which stabilizes *Cpeb4* mRNA and promotes CPEB4 expression during the late phase of the LPS response. Thereafter, ERK1/2 MAPK signaling, which is also activated by LPS, controls the progressive accumulation of the active hyper-phosphorylated form of CPEB4 (*Guillén-Boixet et al., 2016*). Moreover, CPEB4 regulates its own

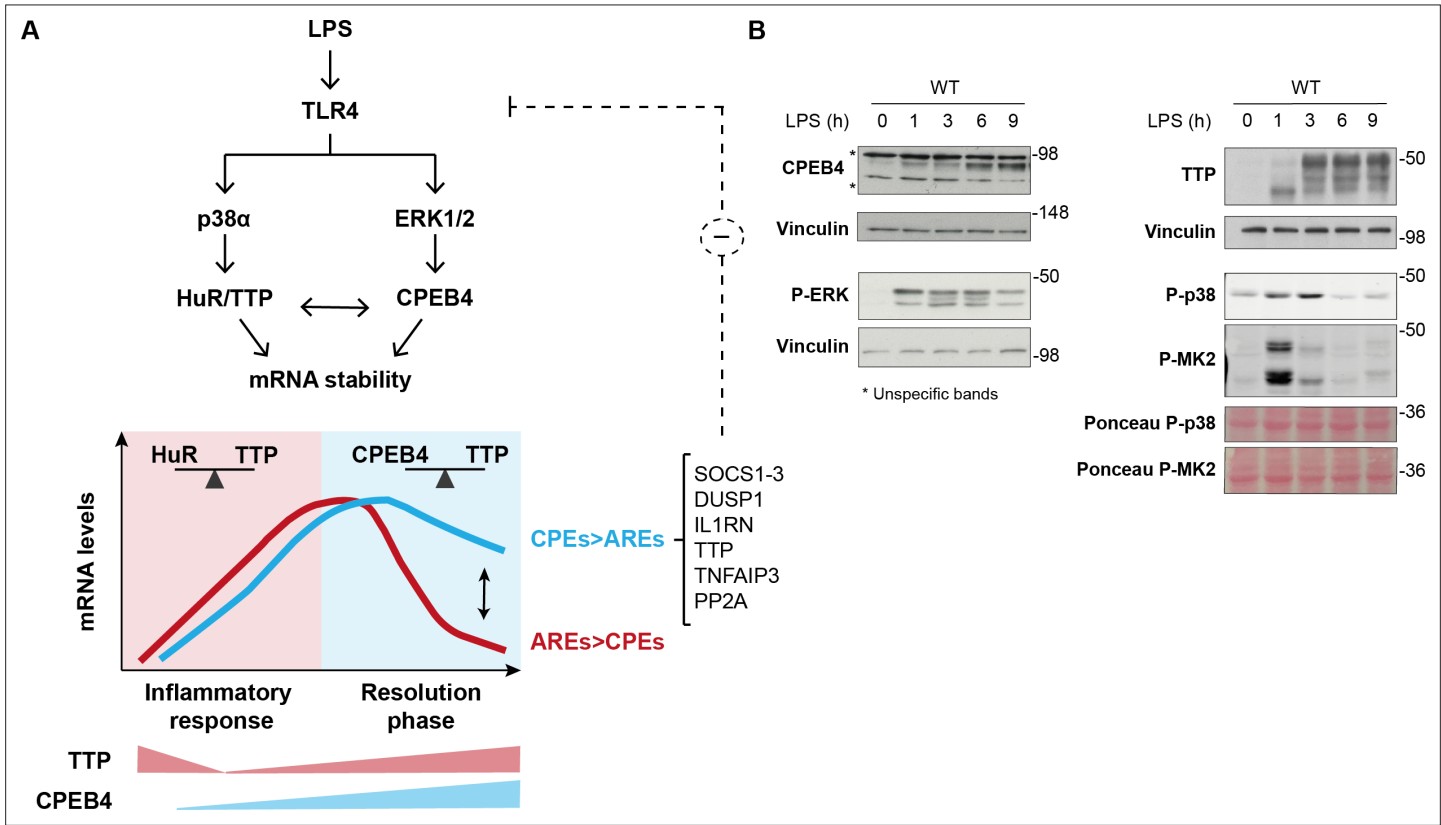

**Figure 6.** Dynamic equilibrium between tristetraprolin (TTP)- and CPEB4-mediated regulation of mRNAs during inflammation resolution. (**A**) Lipopolysaccharide (LPS) stimulates the MAPK signaling cascades downstream of TLR4. p38α controls TTP phosphorylation, causing a shift in the competitive binding equilibrium between Hu-antigen R (HuR) and tristetraprolin (TTP) towards HuR, which stabilizes AU-rich element (ARE)-containing mRNAs. The p38α/HuR/TTP axis also regulates *Cpeb4* mRNA stability and promotes CPEB4 expression during the late phase of the LPS response. CPEB4 then accumulates in its active state, which involves phosphorylation by ERK1/2 MAPK signaling. During the resolutive phase of the LPS response, CPEB4 and TTP compete to stabilize/destabilize mRNAs containing cytoplasmic polyadenylation elements (CPEs) and AREs. The equilibrium between these positive and negative *cis*-acting elements in the mRNA 3'-UTRs generates customized temporal expression profiles. CPEB4 stabilizes CPE-dominant mRNAs, which are enriched in transcripts encoding negative regulators of MAPKs that contribute to inflammation resolution. (**B**) Immunoblot of the indicated proteins in wildtype (WT) BMDMs treated with LPS for 0–9 hr. Vinculin and Ponceau staining served as loading control.

The online version of this article includes the following source data and figure supplement(s) for figure 6:

**Source data 1.** Blots corresponding to *Figure 6B*.

**Figure supplement 1.** Graphical abstract.

mRNA, generating a positive feedback loop that contributes to the increased CPEB4 levels and activity at the late phase of the LPS response. Additionally, CPEB4 binds to the mRNA encoding TTP (*Zfp36*), adding a negative feedback loop that restores TTP activity during the late LPS response. These results highlight the complexity of the superimposed layer of coordination between the MAPKs signaling pathways in regulating levels and activities of both CPEB4 and TTP (*Figure 6A and B*).

This regulatory circuit would generate tight temporal control of the stability of the TTP and CPEB4 co-regulated mRNAs, which is further delineated by the *cis*-acting elements in the 3'-UTR (*Figure 6A and B*). In these transcripts, the equilibrium between CPEs and AREs provides transcript-specific behavior that contributes to uncoupling the expression of pro-inflammatory mediators (which need to be rapidly silenced) from anti-inflammatory mediators (whose synthesis needs to be sustained longer). Accordingly, ARE-d mRNAs are enriched for transcripts that encode pro-inflammatory cytokines, which are expressed during the early phase of the LPS response (*Anderson, 2010*; *Spasic et al., 2012*). In contrast, CPE-d mRNAs are enriched for transcripts that encode factors that contribute to inflammation resolution during the late phase of the LPS response, such as *Socs1*, *Il1rn*, *Dusp1*, and *TTP* itself. Based on these correlations, we propose that the decreased stability of these mRNAs in the absence of CPEB4 could explain the impaired inflammation resolution observed in Cpeb4MKO mice.

CPEB4 targets include negative regulators of MAPKs, which generate a negative feedback loop that limits the extent of the inflammatory response. Accordingly, we found that CPEB4 was over-expressed in patients with sepsis, while myeloid-specific Cpeb4MKO mice with LPS-induced septic shock had lower survival rates and a more exacerbated inflammation than WT mice. Other components of this circuit can also regulate analogous inflammatory phenotypes. Thus, TTP KO mice have exacerbated LPS-induced shock (*Kafasla et al., 2014*), HuR KO mice display inflammatory phenotypes (*Kafasla et al., 2014*), and p38α deletion in macrophages reduces the LPS response and renders mice more resistant to endotoxic shock (*Kang et al., 2008*).

Previous studies have shown that CPEB4's main function is to recruit the polyadenylation machinery and promote the elongation of the poly(A) tail of its targets mRNAs (*Weill et al., 2012*). Thus, the role of CPEB4 in macrophage mRNA stabilization during inflammation suggests that the length of the poly(A) tail is not regulated by an unidirectional process (ARE-mediated deadenylation), but rather as the result of a dynamic equilibrium between cytoplasmic deadenylation and polyadenylation. Moreover, this equilibrium could be modulated, quantitatively and temporarily, by the relative numbers of CPEs and AREs (*Figure 6—figure supplement 1*).

## Materials and methods
### Human sepsis microarray analysis

The GSE65682 (n = 760/42, sepsis patients/healthy persons) and GSE57065 (n = 28/25, sepsis patients/healthy persons) datasets were used. Affymetrix U219 arrays from the GSE65682 cohort were downloaded from NCBI GEO and processed using RMA background correction and summarization with R and Bioconductor (*Gentleman et al., 2004*). Technical variable assessment and adjustment was performed using the Eklund metrics (*Eklund and Szallasi, 2008*), adjusting for RMA IQR, PM IQR, RNA Degradation, and PM Median. Gene expression deconvolution of blood samples to obtain cell proportion estimates was performed using the CellMix package (*Gaujoux and Seoighe, 2013*) with the DSA built-in blood signature database (*Abbas et al., 2009*) and the DSA algorithm with default options (*Figure 1—figure supplement 1*). Initially, differentially expressed genes were determined using limma (*Ritchie et al., 2015*), adjusting by sex and the selected Eklund metrics, and using a |FC| > 2 and a Benjamini–Hochberg adjusted p-value<0.05. For volcano plots, the probeset with highest log2RMA interquartile range (IQR) is considered for each gene; from these selected probesets, those that are identified as myeloid/lymphoid are shown as examples; the y-axis represents the –log10 Benjamini–Hochberg adjusted p-value computed via limma moderated *t*-test (*Figure 1—figure supplement 1*). To assess the consistency of the CPEB4 gene expression differences across groups, regardless of the sample's cell composition, for every cell type we applied a mixed linear model taking the underlying cell-type proportion, the selected Eklund metrics and sex as adjusting covariates, and using the microarray probeset as random effect (*Figure 1—figure supplement 1*). Cpeb4 mRNA expression in human immune cell lineages was obtained from the Haemosphere website (*Choi et al., 2019*; *Figure 1—figure supplement 1*).

## Mouse studies

To generate a myeloid-specific *Cpeb4* KO mice (Cpeb4MKO), conditional *Cpeb4* animals (*Cpeb4*^lox/lox) (*Maillo et al., 2017*) were crossed with *Lyz2*^Cre (*Clausen et al., 1999*) transgenic animals obtained from Jackson Laboratory. Offspring mice were maintained in a C57BL/6J background. Routine genotyping was performed by PCR. BMDMs were obtained from WT or myeloid-specific *Mapk14*KO mice (*Mapk14*^lox/lox*Lyz2*^Cre, referred to as p38αMKO) (*Youssif et al., 2018*) or full *Cpeb4*KO mice (*Cpeb4*^−/−) (*Maillo et al., 2017*). Mice were given free access to food and water and maintained in individually ventilated cages under specific pathogen-free conditions (unless otherwise specified). All experimental protocols were approved by the Animal Ethics Committee at the University of Barcelona. Serum was collected by centrifuging clotted blood at 3000 × *g* for 30 min. Cytokine levels were measured with a murine ProcartaPlex Assay (Labclinics).

## LPS-induced endotoxic shock

WT and *Cpeb4*^ΔM mice were injected intraperitoneally with LPS (10 mg/kg; Santa Cruz SC-3535, *Escherichia coli* 0111:B4). Animals were monitored and samples were collected at the indicated times. Mice between 2 and 5 months of age were used, matched for age and sex.

## Cell culture

BMDMs were isolated from the femurs of adult mice as previously described (*Celada et al., 1996*). Bone marrow cells were differentiated for 7 days on bacteria-grade plastic dishes (Nirco, ref 140298) in DMEM supplemented with 20% (vol/vol) FBS, penicillin (100 units/ml), streptomycin (100 mg/ml), L-glutamine (5 mg/ml), and 20% L-cell conditioned medium as a source of M-CSF. The medium was renewed on day 5 when BMDMs were plated for the experiment. On day 7, media were changed to DMEM containing 10% FBS, penicillin (100 units/ml), streptomycin (100 mg/ml), and L-glutamine (5 mg/ml). On day 8, BMDMs were primed with LPS (10 ng/ml, *E. coli* 0111:B4, Santa Cruz SC-3535) for the indicated time points. To inhibit p38α MAPK, BMDMs were treated with PH797804 (1 µM) for 1 hr prior to LPS treatment. For fluorescence- activated cell sorting (FACS) analysis, a Gallios flow cytometer (Beckman Coulter) was used, and data were analyzed with the FlowJo software. The following antibodies were used: anti-mouse CD11b FITC (BD Biosciences 557396), anti-mouse F4/80 antigen PE (BD Biosciences 12-4801-82), and anti-mouse MHC class II (I-A/I-E) PC (BD Biosciences 17-5321-81).

Osteosarcoma cells (U2OS) were grown as previously reported (*Trempolec et al., 2017*). Briefly, U2OS cells expressing the MKK6 construct were cultured in DMEM containing 10% FBS, penicillin (100 units/ml), streptomycin (100 mg/ml), and L-glutamine (5 mg/ml) in the presence 4 µg/ml Blasticidin S HCl (A11139-03, Invitrogen) and 35 µg/ml Zeocin (R250-01, Invitrogen). To activate the p38α MAPK signaling pathway, cells were treated with 1 µg/ml tetracycline (Sigma 87128-25G) or the corresponding amount of ethanol (1:1000 dilution) as a control. To inhibit p38α, 1 µM PH-797804 (Selleckchem, S2726) or 10 µM SB203580 (Axon Medchem) was used.

## Infection with shHuR-expressing lentivirus

To generate short hairpin RNAs (shRNAs) in a lentivirus delivery system, 293T cells (60% confluent) were transfected with two different shRNA-encoding DNA, pLKO.1-shHuR03 and -shHuR04 (5 µg) (IRB Barcelona Genomic Facility), as well as with VSV-G (0.5 µg) and Δ89 packaging DNA (4.5 µg), using the calcium chloride method. After overnight incubation, the medium was replaced; 48 hr later, medium containing the virus was collected and passed through PVDF filters. For cell infection, virus-containing medium was diluted 1:10 in buffer containing 6 mg/l of polybrene, incubated for 10 min, and then placed on the U2OS cell monolayer. The following day medium was replaced; 24 hr later, cells were incubated with puromycin (2 µg/ml) for a 24- to 48 hr selection. Selected cell populations were cultured in medium with puromycin (2 µg/ml) and used for further analysis. Sequences of shHuR RNA: E03 (CCGGCGTGGATCAGACTACAGGTTTCTCGAGAAACCTGTAGTCTGATCCACGTTTTT); E04 (CCGGGCAGCATTGGTGAAGTTGAATCTCGAGATTCAACTTCACCAATGCTGCTTTTT).

## RNA stability in BMDMs

Cells were stimulated with LPS (10 ng/ml, *E. coli* 0111:B4, Santa Cruz SC-3535) for the indicated times, and control cells (time 0) were collected. Fresh medium and actinomycin D (5–10 µg/ml, Sigma-Aldrich

A9415) were added, and cells were collected at the indicated times. Total RNA was isolated, and cDNA was synthesized for RT-qPCR analysis. At each time point, the remaining mRNA was normalized to *Gapdh* mRNA levels. The value at time 0 was set as 100%, and the percentage of remaining mRNA was calculated for the rest of the time points.

## RNA analysis

Total RNA was either extracted by TRIzol reagent (Invitrogen), using phenol-chloroform, or using Maxwell following the manufacturer's instructions (Maxwell 16 LEV SimplyRNA Cells Kit, Promega). Next, 1 µg of RNA was reverse-transcribed with oligodT and random primers using RevertAid Reverse Transcriptase (Thermo Fisher) or the SuperScript IV First-Strand Synthesis System (Thermo Fisher), following the manufacturer's recommendations. Quantitative real-time PCR was performed in a QuantStudio 6flex (Thermo Fisher) using PowerUp SYBR Green Master (Thermo Fisher). All quantifications were normalized to an endogenous control (*Tbp*, *Rpl0*, *Gapdh*, *18S*). Oligonucleotides used for RT-qPCR are listed in *Supplementary file 7*. Values were normalized by the median expression in untreated WT animals/cells.

## Cell transfection

Murine RAW 264.7 macrophages were cultured in DMEM supplemented with 10% fetal bovine serum, 2 mM L-glutamine, and 1% penicillin/streptomycin at 37°C and 5% $CO_2$. Cells were seeded in p6-well plates at a concentration of 270,000 cells/well; 24 hr later, cells were transiently transfected with 1.5 µg of plasmid using SuperFect Transfection Reagent (QIAGEN ref 301305), following the manufacturer's instructions, for 2.5 hr. Cells were washed twice with PBS and allowed to rest at 37°C for 40 hr. Transfected cells were then treated with LPS (10 ng/ml, *E. coli* 0111:B4, Santa Cruz SC-3535) for the indicated times. For cells used in the stability experiments, actinomycin D (5 µg/ml, Sigma-Aldrich A9415) was added for 0–60 min, and cells were collected to extract total RNA. Firefly was normalized to Renilla, and time 0 of actinomycin D was used as reference for all time points.

## Plasmid construction

To generate a Firefly luciferase reporter mRNA that contained different combinations of CPEs and AREs, we combined two 3'-UTR fragments that were enriched in AREs and CPEs, respectively. Fragments were selected from a CPEB4 and a TTP target mRNA. Short fragments with no other elements were prioritized. For the CPEs fragment, a 67 bp fragment of cyclin B1 3'-UTR containing three CPEs was selected. For the AREs fragment, a 132 bp fragment at the 3'-UTR of immediate early response 3 (Ier3) mRNA was selected. Wildtype Ier3 fragment contained five AREs, and two modified fragments with three or five AREs mutated were designed (two AREs and zero AREs, respectively) (*Supplementary file 8*). Firefly luciferase gene with downstream Ier3 ARE fragments were purchased from GeneArt (Thermo Fisher). The three fragments containing zero, two, or five AREs were cloned with BamHI and HindIII restriction enzymes in pCMV-lucRenilla plasmid (*Villanueva et al., 2017*) upstream of the 3'-UTR of cyclin B1 mRNA. Mutation of the three CPEs of cyclin B1 3'-UTR (5'-TTTTAAT-3' → 5'-TTgggAT-3'; 5'-TTTTACT-3' → 5'-TTggACT-3'; 5'-TTTAAT-3' → 5'-TTggAAT-3') was done using the QuikChange Lightning Multi Site-Directed Mutagenesis Kit (Agilent).

## Immunoblotting

Protein extracts were quantified by DC Protein assay (Bio-Rad), and equal amounts of proteins were separated by SDS-polyacrylamide gel electrophoresis. After transfer of the proteins onto a nitrocellulose membrane (GE10600001, Sigma) for 1 hr at 100 mV, membranes were blocked in 5% milk, and specific proteins were labeled with the following antibodies: CPEB4 (Abcam Ab83009/clone 149C/D10, monoclonal homemade); HuR (3A2, sc-5261 Santa Cruz); HIF1a (Cayman 10006421); phospho-p44/42 (Erk1/2) (Thr202/Try204) (Cell Signaling clone E10, 9106); SOCS1 (Abcam Ab9870); phospho-p38 (Thr180/Y182) (Cell Signaling, 9211S); p38α (C-20)-G (Santa Cruz, sc-535-G); phospho-MAPKAPK2 (Thr222) (Cell Signaling, 3044S), TTP (D1I3T) (Cell Signaling, 71632), and vinculin (Abcam Ab18058). Quantification was done with ImageStudioLite software, and protein expression was normalized by the loading control signal.

## Lambda protein phosphatase assay (λ-PPase)

BMDMs were lysed in l-PPase reaction buffer (New England Biolabs, Ipswich, MA) supplemented with 0.4% NP-40 and EDTA-free protease inhibitors (Sigma-Aldrich). $\lambda$-PPase (New England Biolabs) reactions were performed following the manufacturer's instructions.

## RNAseq and data analysis: Preprocessing and normalization

Reads were aligned to the mm10 genome using STAR 2.3.0e (*Dobin et al., 2013*) with default parameters. Counts per genomic feature were computed with the function featureCounts R (*Liao et al., 2014*) of the package Rsubread (*Liao et al., 2013*). Counts data were normalized using the rlog function of the DESeq2 R package (*Love et al., 2014*). Sample information, alignment statistics, and principal component analysis are shown in *Supplementary file 1*.

## Differential expression at gene level

Differential expression analyses between genotype conditions (*Cpeb4*$^{-/-}$ vs. WT) were performed using the DESeq2 R package (*Love et al., 2014*) on the original counts data. This was done independently for every LPS time point. To study expression patterns over time for WT samples, the average was taken from the back-transformed (over the four samples) rlog expression to obtain normalized counts. These were further scaled by the maximum value obtained throughout the LPS response for each gene.

## Gene set analysis

Genes were annotated according to Hallmark terms (*Liberzon et al., 2015*) using the R package *org.Hs.eg.db* (*Carlson, 2019*). Gene set analysis was then performed using the regularized log transformed matrix, in which genes with low expression (genes with less than an average count of five reads) were filtered out. The rotation-based approach for enrichment (*Wu et al., 2010*) implemented in the R package limma (*Ritchie et al., 2015*) was used to represent the null distribution. The maxmean enrichment statistic proposed in *Efron and Tibshirani, 2007*, under restandardization, was considered for competitive testing.

## Visualization

To obtain a measure of the pathway activity in the transcriptomic data, we summarized each Hallmark as z-score gene signature (*Lee et al., 2008*). To this end, rlog normalized expression values were centered and scaled gene-wise according to the mean and standard deviation computed across samples, which were then averaged across all genes included in that Hallmark term. In addition, a global signature was computed for all the genes in the expression matrix and then used for a priori centering of Hallmark scores. This strategy can avoid systematic biases caused by the gene-correlation structure present in the data and can adjust for the expectation under gene randomization, that is, the association expected for a signature whose genes have been chosen at random (*Efron and Tibshirani, 2007*; *Lee et al., 2008*). All statistical analyses were carried out using R and Bioconductor (*Huber et al., 2015*).

## RNA-immunoprecipitation (RIP)

*Cpeb4*$^{-/-}$ and WT primary BMDMs, untreated or stimulated for 9 hr with LPS (10 ng/ml), were cultured in DMEM supplemented with 10% FBS, rinsed twice with 10 ml PBS, and incubated with PBS 0.5% formaldehyde for 5 min at room temperature under constant soft agitation to crosslink RNA-binding proteins to target RNAs. The crosslinking reaction was quenched by addition of glycine to a final concentration of 0.25 M for 5 min. Cells were washed twice with 10 ml PBS, lysed with a scraper and RIPA buffer (25 mM Tris-Cl pH 7.6, 1% Nonidet P-40, 1% sodium deoxycholate, 0.1% SDS, 100 mM EDTA, 150 mM NaCl, protease inhibitor cocktail, RNase inhibitors), and sonicated for 10 min at low intensity with Standard Bioruptor Diagenode. After centrifugation (10 min, max speed, 4°C),

supernatants were collected, precleared, and immunoprecipitated (4 hr, 4°C, on rotation) with 10 µg anti-CPEB4 antibody (Abcam), and 50 µl Dynabeads Protein A (Invitrogen). Beads were washed four times with cold RIPA buffer supplemented with protease inhibitors (Sigma-Aldrich), resuspended in 100 µl Proteinase-K buffer with 70 µg Proteinase-K (Roche), and incubated for 60 min at 65°C. RNA was extracted by standard phenol-chloroform, followed by Turbo DNA-free Kit (Ambion) treatment.

## Sequencing and analysis

Samples were processed at the IRB Barcelona Functional Genomics Facility following standard procedures. Libraries were sequenced by Illumina 100 bp single-end and FastQ files for Input and IP samples were aligned against the mouse reference genome mm10 with Bowtie2 version 2.2.2 (*Langmead and Salzberg, 2012*) in local mode accepting one mismatch in the read seed and using default options. Alignments were sorted and indexed with sambamba v0.5.1 (*Tarasov et al., 2015*). Putative over-amplification artifacts (duplicated reads) were assessed and removed with the same software. Coverage tracks in TDF format for IGV were made using IGVTools2 (*Tarasov et al., 2015*). Quality control assessment for unaligned and aligned reads was done with FastQC 0.11 (*Brown et al., 2017*). Further quality control steps (Gini/Lorenz/SSD IP enrichment, PCA) were performed in R with the htSeqTools package version 1.12.0 (*Planet et al., 2012*). rRNA contamination was assessed using the mm10 rRNA information available in the UCSC genome browser tables. A preliminary peak calling process was performed for duplicate filtered sequences with the MACS 1.4.2 software (*Zhang et al., 2008*) using default options to assess overall enrichment in the IP sequenced samples against their respective Input controls. The identified peaks were annotated against the mouse reference genome mm10 annotation using the ChIPpeakAnno package version 2.16 (*Zhu et al., 2010*). Additionally, quantification for mm10 UCSC 3'-UTR genomic regions (Biomart ENSEMBL archive February 2014) in all samples was performed using the countOverlaps function from the GenomicRanges package version 1.34.0, and potentially enriched 3'-UTRs between WT and Cpeb4–/– IPs were identified using the enrichedRegions function from the htSeqTools package version 1.30.0 using default options and reporting Benjamini–Yekutieli adjusted p-values (pvBY). For each analyzed 3'-UTR genomic region, the maximum peak score for overlapping MACS peaks was retrieved (MaxScore). 3'-UTR regions with no overlapping MACS reported peaks were given a MaxScore value of 0. Finally, CPEB4-associated mRNAs were defined as those with either (1) pvBY < 0.02 and $\log2_{RPKM}FC > 0.5$; (2) pv < 0.02 and $\log2_{RPKM}FC > 1.2$; or (3) MaxScore > 500 and $\log2_{RPKM}FC > 0.5$. Gene Ontology analysis was performed using the DAVID Functional Annotation Bioinformatics Microarray Analysis, version 6.8 (*Dennis et al., 2003*).

## RIP and RT-qPCR

RIP was performed as for RIP-seq except that the obtained RNA was reverse-transcribed and analyzed by RT-qPCR (see RNA analysis). The following antibodies were used: mouse monoclonal CPEB4 (ab83009 Lot. GR95787-2, Abcam); HuR antibody (3A2, sc-5261 SantaCruz); and normal mouse IgG polyclonal antibody (12–371 Sigma). Protein A/G beads were used in each case (Invitrogen).

## CPE-containing mRNAs

To analyze CPE-A-containing mRNAs, the script developed by Piqué et al. was run over mm10 3'-UTR reference sequences (Biomart ENSEMBL archive February 2014). mRNAs containing a putative 3'-UTR with a CPE-mediated repression and/or activation prediction were considered CPE-containing mRNAs. For CPE-G-containing mRNAs, the same script was adapted to mRNAs containing a putative 3'-UTR with the TTTTGT motif within the optimal distances to the polyadenylation signal (PAS) established by *Piqué et al., 2008* (*Supplementary file 9*). As a control for the enrichment in CPE-containing mRNAs in CPEB4 target lists, the BMDM transcriptome was defined as those transcripts with >10 reads in the corresponding WT input (untreated/LPS-stimulated BMDMs).

## ARE-containing mRNAs

For each gene, the reference sequence of the longest 3'-UTR was selected. The number of AREs was calculated by scanning the corresponding 3'-UTR and counting the number of occurrences of the ARE motif (AUUUA) and overlapping versions (AUUUAUUUA) up to five overlapping motifs. Note that k overlapping motifs count as k ARE motif. ARE-containing mRNAs were considered as those with two or more AUUUA motifs (*Gruber et al., 2011*).

## ARE:CPE score

For computation of the ARE:CPE score, the reference sequence of the longest 3'-UTR per gene was selected. Similarly to the number of AREs, the number of CPEs was calculated by scanning the corresponding 3'-UTR and counting the number of occurrences of all the CPE motifs ('TTTTAT,' 'TTTTAAT,' 'TTTTAAAT,' 'TTTTACT,' 'TTTTCAT,' 'TTTTAAGT,' 'TTTTGT'). A version without considering the last motif yielded no significant differences. The optimal distances to the PAS established by *Piqué et al., 2008* were not considered for this analysis.

## TTP and HuR target mRNAs

Data were obtained from TTP and HuR PAR-iCLIP experiments in LPS-stimulated BMDMs (*Sedlyarov et al., 2016*). For TTP, mRNAs bound after 3 hr or 6 hr of LPS stimulation were considered. For HuR-bound mRNAs, PAR-iCLIP data corresponded to 6 hr of LPS stimulation. Only mRNAs with HuR/TTP binding in the 3'-UTR were considered.

## Overlap between TTP and CPEB4 target mRNAs

CPEB4 targets in LPS-stimulated macrophages were defined as stated in the RIP section. For TTP, mRNAs bound after 3 hr or 6 hr of LPS stimulation in Sedlyarov datasets were considered (*Sedlyarov et al., 2016*). The BMDM transcriptome was defined as those transcripts with >10 reads in the LPS-stimulated WT input from the RIPseq experiment. The percentage of CPEB4 target mRNAs in BMDMs transcriptome and TTP targets was calculated.

## ARE- and CPE-containing mRNAs

The list of 1521 ARE- and CPE-containing mRNAs contained HuR, TTP, or CPEB4 target mRNAs that had at least one ARE motif and one CPE motif in their 3'-UTR. mRNAs containing only AREs or only CPEs were excluded.

## mRNA levels/stability in TTP$^{\Delta M}$ BMDMs

Data were obtained from *Sedlyarov et al., 2016*. The statistical analysis of the publication was also considered.

## Statistics

Data are expressed as mean ± SEM (beingSEM = SD/√n). Dataset statistics were analyzed using the GraphPad Prism Software. For two group comparisons, column statistics were calculated and based on standard deviation, D'Agostino–Pearson normality test, parametric *t*-test (assuming or not same SD), or nonparametric *t*-test (Mann–Whitney) was performed. For multiple comparisons, one-way ANOVA Kruskal–Wallis test or two-way ANOVA followed by the Bonferroni post-hoc test was used. To assess mouse survival, Kaplan–Meier survival curves were computed with R2.15 and the survival package version 2.37-2 (*Piqué et al., 2008*) using the likelihood-ratio test, adjusting by gender, weight, and LPS dose. Fisher's exact test was used for contingency analysis. Differences under $p < 0.05$ were considered statistically significant (\*$p < 0.05$, \*\*$p < 0.01$, \*\*\*$p < 0.001$; \*\*\*\*$p < 0.0001$). Differences under $p < 0.1$ are indicated as ns (nonsignificant). For animal studies, mice were grouped in blocks to control for experimental variability. Furthermore, WT and KO mice were bred in the

same cages whenever possible, and the experimenter was blinded until completion of the experimental analysis. Experiments were repeated independently with similar results, as indicated in the figure legends.

## Acknowledgements

We thank the Biostatistics/Bioinformatics, Histopathology, Mouse Mutant, and Functional Genomics facilities at IRB Barcelona. The Flow Cytometry Facility of the UB/PCB and the CRG Genomic Unit are also acknowledged. We thank Dr. Mercedes Fernández and members of the labs of Dr. Angel Nebreda and Dr. Raul Méndez for useful discussion. This work was supported by grants from the Spanish Ministry of Economy and Competitiveness (MINECO, BFU2017-83561-P), the Fundación BBVA, the Fundación Bancaria 'la Caixa,' the Fundació La Marató TV3, and the Scientific Foundation of the Spanish Association Against Cancer (AECC). CS is the recipient of an FPI-Severo Ochoa fellowship from MINECO. IRB Barcelona is the recipient of a Severo Ochoa Award of Excellence from MINECO (Government of Spain) and was supported by the CERCA Programme (Catalan Government).

## Additional information

### Funding

| Funder | Grant reference number | Author |
|---|---|---|
| Ministerio de Economía y Competitividad | BFU2017-83561-P | Raúl Méndez |
| BBVA Foundation | | Raúl Méndez |
| "la Caixa" Foundation | | Raúl Méndez |
| Fundación Científica Asociación Española Contra el Cáncer | | Raúl Méndez |
| Ministerio de Economía y Competitividad | Contrato Predoctoral Severo Ochoa | Clara Suñer |
| Marie Curie | COFOUND IRB PostPro 2.0 | Annarita Sibilio |
| Fundació la Marató de TV3 | | Raúl Méndez |

The funders had no role in study design, data collection and interpretation, or the decision to submit the work for publication.

### Author contributions

Clara Suñer, Conceptualization, Data curation, Formal analysis, Investigation, Methodology, Project administration, Supervision, Visualization, Writing – original draft, Writing – review and editing; Annarita Sibilio, Conceptualization, Data curation, Formal analysis, Investigation, Methodology, Project administration, Writing – original draft, Writing – review and editing; Judit Martín, Chiara Lara Castellazzi, Ivan Dotu, Adrià Caballé, Elisa Rivas, Vittorio Calderone, Data curation, Formal analysis, Investigation, Methodology, Writing – review and editing; Oscar Reina, Data curation, Formal analysis, Methodology, Writing – review and editing; Juana Díez, Funding acquisition, Project administration, Supervision, Writing – review and editing; Angel R Nebreda, Conceptualization, Funding acquisition, Investigation, Methodology, Project administration, Supervision, Writing – original draft, Writing – review and editing; Raúl Méndez, Conceptualization, Funding acquisition, Methodology, Project administration, Resources, Supervision, Writing – original draft, Writing – review and editing

### Author ORCIDs

Clara Suñer (iD) http://orcid.org/0000-0003-2581-8864
Angel R Nebreda (iD) http://orcid.org/0000-0002-7631-4060
Raúl Méndez (iD) http://orcid.org/0000-0002-1952-6905

## Ethics

This study was performed in strict accordance with the recommendations of the European Directive 2010/63/EU on the protection of animals used for scientific purposes. All experimental protocols were approved by the Animal Ethics Committee at the Parc Cientific de Barcelona.

## Decision letter and Author response

Decision letter https://doi.org/10.7554/eLife.75873.sa1
Author response https://doi.org/10.7554/eLife.75873.sa2

---

# Additional files

## Supplementary files

• Supplementary file 1. RNAseq wildtype_vs_*Cpeb4* KO bone marrow-derived macrophages (BMDMs). Wildtype and *Cpeb4* KO BMDMs were stimulated with lipopolysaccharide (LPS) and mRNA levels were quantified by RNAseq (n = 4). Differential expression between genotype conditions was performed with DESeq2 R package. Wildtype and *Cpeb4* KO samples were compared for each time point independently. Sample information, alignment statistics, and principal component analysis are shown.

• Supplementary file 2. RIPseq-defined CPEB4 target mRNAs. Wildtype and *Cpeb4* KO bone marrow-derived macrophages (BMDMs) were left untreated or stimulated with lipopolysaccharide (LPS) for 9 hr. Immunoprecipitation (IP) with anti-CPEB4 antibody was then performed, and RNA was extracted and analyzed by RNAseq. CPEB4 targets were defined based on the enrichment between wildtype and *Cpeb4* KO IPs.

• Supplementary file 3. Genome-wide AU-rich elements (AREs), cytoplasmic polyadenylation elements (CPEs), and ARE/CPE score. For each gene, the reference sequence of the longest 3′-UTR was selected. The number of AREs and CPEs was calculated by scanning the corresponding 3′-UTR and counting the number of occurrences of each motif. The ARE/CPE score was calculated as the log2 transformed ratio between the number of ARE and CPE motifs.

• Supplementary file 4. Tristetraprolin (TTP) and Hu-antigen R (HuR) target mRNAs. Data were obtained from TTP and HuR PAR-iCLIP experiments in lipopolysaccharide (LPS)-stimulated bone marrow-derived macrophages (BMDMs) (*Sedlyarov et al., 2016*). Only mRNAs with HuR/TTP binding in the 3′-UTR were considered. For TTP, mRNAs bound after 3 hr or 6 hr of LPS stimulation were considered. For HuR-bound mRNAs, PAR-iCLIP data corresponded to 6 hr of LPS stimulation.

• Supplementary file 5. RNAseq wildtype bone marrow-derived macrophages (BMDMs). Wildtype BMDMs were stimulated with lipopolysaccharide (LPS), and mRNA levels were quantified by RNAseq (n = 4). Wildtype and *Cpeb4* KO samples were compared for each time point independently. The expression pattern over time for wildtype samples was analyzed (differential expression results against consecutive time points and Rlog data normalized by maximum).

• Supplementary file 6. AU-rich element (ARE)- and cytoplasmic polyadenylation element (CPE)-containing mRNAs. ARE- and CPE-containing mRNAs were defined as all mRNAs regulated by CPEB4, tristetraprolin (TTP), or Hu-antigen R (HuR) in lipopolysaccharide (LPS)-stimulated bone marrow-derived macrophages (BMDMs) (see also *Supplementary files 2 and 4*).

• Supplementary file 7. Primers used for RT-qPCR analysis.

• Supplementary file 8. Constructs.

• Supplementary file 9. Scripts for CPE-A and CPE-G. To analyze CPE-A-containing mRNAs, the script developed by Piqué et al. was run over mm10 3′-UTR reference sequences (Biomart ENSEMBL archive February 2014). mRNAs containing a putative 3′-UTR with a cytoplasmic polyadenylation element (CPE)-mediated repression and/or activation prediction were considered CPE-containing mRNAs. For CPE-G-containing mRNAs, the same script was adapted to mRNAs containing a putative 3′-UTR with the TTTTGT motif within the optimal distances to the polyadenylation signal (PAS) established by *Piqué et al., 2008*.The original and modified script, as well as the input and output files of the CPE-A and CPE-G analysis, are included.

• Transparent reporting form

## Data availability

Raw data for RIP-seq and RNA-seq datasets are available in GEO (accession number GSE160191 and GSE160246, respectively). Numerical data from genome-wide experiments and motif analysis

are available in supplementary tables 1-6. All blots shown and used for quantifications have been provided as source data. Scripts are available as Supplementary file 9.

The following datasets were generated:

| Author(s) | Year | Dataset title | Dataset URL | Database and Identifier |
|---|---|---|---|---|
| Suñer C, Mendez R | 2022 | Macrophage inflammation resolution requires CPEB4-directed offsetting of mRNA degradation | http://www.ncbi.nlm.nih.gov/geo/query/acc.cgi?acc=GSE160191 | NCBI Gene Expression Omnibus, GSE160191 |
| Suñer C, Mendez R | 2022 | CPEB4-mediated translational regulation of LPS response in Bone Marrow Derived Macrophages (BMDMs) | http://www.ncbi.nlm.nih.gov/geo/query/acc.cgi?acc=GSE160246 | NCBI Gene Expression Omnibus, GSE160246 |

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
