## [Editor Report]

This article by Suner et al. investigated specific regulation of LPS-induced sepsis and mechanisms underlying resolution of inflammation. The authors focused on CPEB4 and TTP, two RNA-binding proteins, showing that they work in opposite manner to regulate RNA stability in macrophages and modulate inflammation. This is an interesting study that offers significant advance to the fields of sepsis-inflammation and post-transcriptional regulation of gene expression in inflammatory responses.

---

## [Decision Letter]

**Decision letter after peer review:**

Thank you for submitting your article "Inflammation resolution requires CPEB4-directed offsetting of mRNA degradation" for consideration by *eLife*. Your article has been reviewed by 2 peer reviewers, and the evaluation has been overseen by a Reviewing Editor and Mone Zaidi as the Senior Editor. The reviewers have opted to remain anonymous.

Essential revisions:

Please address all comments raised by both reviewers in the public evaluation summaries and recommendations for authors sections below

*Reviewer #1 (Recommendations for the authors):*

1. In relation to figure S1B it is argued that monocytes are the immune cell population expressing the highest Cpeb4 mRNA level, but the figure indicates that eosinophil lineage cells express as much.

2. It is unclear why Il6, Tnf and Il1a were selected for studies in figure 1D.

3. In figure S2E it is unclear what the different stars are indicating. E.g. is the difference for negative regulation of inflam. response not significant?

4. A negative control transcript would be a good addition to figure 3C.

5. Line 136. It is argued that Cpeb4 mRNA was not enriched in p38alphaM cells, but it clearly is, albeit not to the same extent.

6. It is claimed that Cxcl2 shows and intermediate ARE/CPE ratio similar to Ccl2 but in figure 5F it looks more similar to Cxcl1 and Ptgs2.

7. Does the 102/193 TTP bound mRNAs that were also co-IP with CPEB4 represent an enrichment?

8. Line 301, should be "adjusted p-value" and not "p-value".

9. The deconvolution parts need to be clarified.

10. I might have missed a table somewhere, but I did not find information about primers used in RT-PCR nor sequence of shRNA.

11. Line 379 "ler3”?

12. Line 397-400. Were these methods used?

13. I would recommend adding basic information about the RNA sequencing experiments including sequencing depth, number of aligned reads etc. Adding basic QC including PCA plots or similar to show that samples segregate based on condition would strengthen the conclusions relating to the RNA sequencing experiment.

14. Statistics for RNAseq seems to be done independently per time point. How were the resulting p-values adjusted? It seems that all p-values from all time-points should be adjusted together. Would it nor be of interest to use models that fit data across multiple time points?

15. Is formaldehyde required for RIP with CPEB4?

16. When using DAVID, what version of the GO database was used?

17. Why was the February 2014 3'UTR sequences used rather than a more up-to-date version?

18. SEM are used as error bars when plotting data while SD seems more appropriate.

19. Converting the Socs1 mRNA to show an LPS-response similar to Ptgs2 by mutating CPEs in the Socs1 3'UTR would strengthen the conclusion regarding the hypothesis of the relative number of AREs vs CPEs.

*Reviewer #2 (Recommendations for the authors):*

This is a thorough manuscript with an impressive scope of experiments that the authors should be very proud of.

Specific Comments:

(a) Line 95. The authors write: "Cpeb4(deltaM) animals also presented splenomegaly (Figure 1C) and increased splenic levels of the cytokines Il6, Tnf, and Il1a (Figure 1D)." Il6 and Tnf appear to have robust and reproducible effects, but the Il1a effect appears non-significant.

(b) Line 115. The authors write: "…the levels of anti-inflammatory transcripts, such as Il10 mRNA, were reduced in the Cpeb4 -/- BMDMs (Figure 2H)". Is this really a robust and measurable effect? Error bars on these measurements appear to be overlapping at all timepoints.

(c) Line 135. The authors write: "We observed HuR binding to Cpeb4 mRNA was strongly enriched after LPS treatment in WT BMDMs but not in p38a(deltaM) BMDMs…(Figures 3D, S3C)." On the contrary, from Figure 3D, it looks to me like there is actually quite strong Cpeb4 enrichment in HuR IPs upon LPS treatment for both WT and p38a(deltaM) BMDMs; untreated WT Cpbe4 looks to be ~0.015 and treated WT ~1.0 (~60-fold enrichment), whereas untreated p38a(deltaM) is very small, well less than 0.005 presumably, and treated p38a(deltaM) is ~0.4 (~80-fold enrichment). Regardless of the exact numbers, there appears to be robust enrichment of Cpeb4 mRNA in both the WT and p38a(deltaM) macrophages upon LPS-treatment.

(d) Line 142. The authors write: "…significantly reduced decay rates of both Cpeb4 and Tnf mRNAs in macrophages from mice with myeloid cell-specific TTP deletion" However, from Figure 3F, it looks like there is a large effect for Tnf but actually a much smaller one for Cpeb4. Are both of these decay rates significantly reduced? What are the errors and significance on the RNA decay rates shown in Figure 3F?

(e) I like the luciferase-based reporter assays in Figures 4J and 5K used to demonstrate that the number of CPEs directly influences mRNA stability here; in both cases, definitively linking this to CPEB4 binding (as opposed to other CPEBs or an unknown RBP, for example), would be significantly strengthened by looking for the absence of this affect in a Cpeb4 -/- background. Have the authors considered undertaking these control experiments?

(f) The last sentence of the manuscript suggests that mRNA stabilization during inflammation results from a dynamic equilibrium between cytoplasmic deadenylation and polyadenylation. This statement is I assume based on the characterized/known function of CPEs and CPEB4 on polyA tails, but there's no data in the current manuscript on polyA tail length changes in response to CPEB4 binding during inflammation in macrophages. Has this been previously characterized in a similar context? Is it a certainty that the CPEB4-dependent effects on mRNA stability measured in this manuscript result from changes in polyA tail length?

I think my comments, concerns, and questions listed above are self-explanatory, so I don't have anything additional to add along those lines. This is a dense paper, and I am not a technical expert in every area covered, so forgive me if I missed anything!

The one additional comment I have is on whether *eLife* is the right home for this manuscript:

This paper builds from previous work from the authors and other labs that show how CPEB4 and TTP binding to internal elements (CPEs and AREs) can perturb RNA stability in a transcript-specific manner; this is similar to the biochemistry of many other well-characterized RNA element-binding proteins and their effects on stability, which has been described for many different proteins, transcripts, and biochemical pathways. This work provides a new example for how two RNA-element binding proteins, CPEB4 and TTP, regulate mRNA stability by binding their target elements in mRNAs during inflammation. While this is no doubt interesting and important during inflammation, because there are many similar effects that have been reported for different RNA-binding proteins impacting RNA stability in many different contexts, I'm skeptical the findings are of broad interest to the readership of *eLife*; I see this story as one of more specific interest to those in the RNA decay and/or inflammation communities.

---

## [Author Response]

Reviewer #1 (Recommendations for the authors):1. In relation to figure S1B it is argued that monocytes are the immune cell population expressing the highest Cpeb4 mRNA level, but the figure indicates that eosinophil lineage cells express as much.

We thank the reviewer for drawing our attention to this incorrect wording. We have modified the text accordingly. We highlighted only the Monocytes/macrophages because they change in sepsis.

The text now reads:

“The peripheral blood of sepsis patients displays an increased number of monocytes/macrophages, which are one of the immune cell populations that express higher Cpeb4 mRNA levels (Figure 1 —figure supplement 1).”

2. It is unclear why Il6, Tnf and Il1a were selected for studies in figure 1D.

The selection of these genes was based on septic shock literature, where these cytokines have been extensively used as a readout of the induction and resolution of the inflammatory process. We have clarified this point with relevant literature citation in the text (Schulte, et al., 2013).

3. In figure S2E it is unclear what the different stars are indicating. E.g. is the difference for negative regulation of inflam. response not significant?

Asterisks always indicate significant results (*p < 0.05, **p < 0.01, ***p < 0.001; ****p < 0.0001), as stated in the Materials and methods. Previously, in Figure S2E, we indicated that the difference for Negative regulation of inflammatory response was not significant, even if we observed a consistent trend. As mentioned above, we have now replaced these two panels by the GSEA descriptive information (now Figure 2 —figure supplement 3).

4. A negative control transcript would be a good addition to figure 3C.

We have included the comparison (ratio) 18S/GAPDH as Figure 3 —figure supplement 2. showing no significant differences between the WT and KO.

5. Line 136. It is argued that Cpeb4 mRNA was not enriched in p38alphaM cells, but it clearly is, albeit not to the same extent.

We have corrected the text indicating that:

“We observed that HuR binding to Cpeb4 mRNA was strongly enriched after LPS treatment. In p38aMKO BMDMs, the binding of HuR to Cpeb4 and tnf mRNAs was significantly reduced compared to WT BMDMs (Figures 3D, Figure 3 —figure supplement 3)”.

Please note that the levels without treatment are so close to the background that is not possible to make a direct ratio +/- LPS.

6. It is claimed that Cxcl2 shows and intermediate ARE/CPE ratio similar to Ccl2 but in figure 5F it looks more similar to Cxcl1 and Ptgs2.

We apologize for not being clear in our explanation. We wanted to indicate that the two genes, despite having both CPEs and ARES, had opposite ARE/CPE enrichments. We have clarified this point, highlighting the that Cxcl2 3’UTR contains both CPEs and AREs, with a more dominant function of the latter (ARE-dominant 3’UTR):

*“*the 3′-UTRs of *Socs1* and *Il1rn* mRNAs were enriched in CPEs, those of *Cxcl1* and *Ptgs2* mRNA were enriched in AREs, and those of *Ccl2* and *Cxcl2* mRNA presented both but with opposite ARE/CPE ratios.”

7. Does the 102/193 TTP bound mRNAs that were also co-IP with CPEB4 represent an enrichment?

By RIP-seq analysis, we identified 1,173 and 1,829 CPEB4-associated mRNAs in untreated and LPS-treated BMDMs, respectively. Input samples contained 12,874 and 14,308 mRNAs, in untreated and LPS-treated BMDMs, respectively. Thus, CPEB4-associated mRNAs accounted for up to 9% and 13% of the transcriptome in untreated and LPS-treated BMDMs. When we considered TTP targets, CPEB4-bound mRNAs represented 53% of the dataset. Thus, there is around a 5-fold enrichment. We have modified Figure 5B to show these ratios.

8. Line 301, should be "adjusted p-value" and not "p-value".

We thank the reviewer for pointing out this error. We have corrected the sentence accordingly.

9. The deconvolution parts need to be clarified.

We have extended the Materials and methods section to clarify how the deconvolution was performed.

10. I might have missed a table somewhere, but I did not find information about primers used in RT-PCR nor sequence of shRNA.

The sequence for the primers used is now indicated in Supplementary file 7. We apologize for not indicating the sequence of shRNA. It has now been included in the Materials and methods. We have also included a new Supplementary file 8 with the sequences of the CPE and CPE/ARE constructs.

11. Line 379 "ler3”?

Ier3 stands for Immediate Early Response 3 gene, a TTP-regulated mRNA during macrophage inflammatory response to LPS (Sedlyarov et al., 2016). Its 3’UTR contains five AREs and no CPEs in a 130 bp fragment. Based on these characteristics, it was selected to generate the CPE/ARE constructs. We have given this information in the Materials and methods and included the sequences of the constructs in Supplementary file 8.

12. Line 397-400. Were these methods used?

Lambda protein phosphatase assay was performed to confirm that the doublet observe in CPEB4 immunoblots corresponded to the phosphorylated form of CPEB4 (Figure 2C).

13. I would recommend adding basic information about the RNA sequencing experiments including sequencing depth, number of aligned reads etc. Adding basic QC including PCA plots or similar to show that samples segregate based on condition would strengthen the conclusions relating to the RNA sequencing experiment.

We have added this information in Supplementary file 1. We have now included the sample information and alignment statistics obtained with STAR. A supplementary figure with the PCA representation is included in the same file. While first and second components, explaining 74% of the data variance, separate the five time points, the third and fourth components (10% of the variance) distinguish between the KO and WT samples.

14. Statistics for RNAseq seems to be done independently per time point. How were the resulting p-values adjusted?

The p-values were adjusted gene/geneset-wise independently for each time point, and we have now made this clear in the Methods section.

Although we agree that p-values could also be corrected across the different time points to prevent false positives, after discussing this thoroughly, we decided against it for two reasons.

The first consideration is related to the philosophy and purpose of these types of experiments; we understand that techniques such as microarrays or RNAseq that screen the biological system of interest in the whole genome were designed as tools for hypothesis generation and for evidence support but were never meant for validation purposes. For instance, in our experiments, they served to complement evidence from other independent studies and were mostly used to select candidate genes from particular pathways. The selected genes could then be validated in independent data with the statistical power being less compromised by multiple testing corrections. With the current p-value adjustment, we control the false discovery rate genome-wide* for every time point even though we know that a great part of the genome might remain unaltered under the condition of interest. We understand that, in every gene singled out from the whole in our experiment, we are already being quite conservative by adjusting with the tests for the rest of the genome.

The second consideration is that, a priori, as we already anticipated quite different behaviors for Cpeb4-KO activity depending on the LPS exposure time, we expected the changes to occur at the end of the time course. Therefore, we did not want to compromise the statistical power at later time points from genome-wide differential expression comparisons at initial time points.

It seems that all p-values from all time-points should be adjusted together. Would it nor be of interest to use models that fit data across multiple time points?

This is something we attempted when analyzing the RNAseq data but eventually discarded.

We considered the longitudinal model proposed by Fischer et al., [1], which is implemented in the *impulseDE2* R package. For a sufficient number of time points, this method proved more powerful in finding differential expression genes than models that use time as a discrete variable. They claimed that to avoid over-fitting, there should be at least as many time points as number of degrees of freedom used in the model, which is a total of six by ImpulseDE2. Indeed, Spies et al., [2] performed a comparative analysis for time-course differential expression methods and found that classical differential expression with deseq2/edgeR with naïve pairwise comparisons outperformed approaches based on time-courses such as impulseDE2 when the time point size was inferior to 8.

Nevertheless, we believed that the impulseDE2 time-course model could still be useful for monotonous increasing/decreasing behaviors over time. Therefore, we attempted to fit impulseDE2 models in our data but observed patterns that were difficult to interpret. Below we show two genes that were identified as significant by impulseDE2, in which curve fits were really hard to justify under visual inspection.

15. Is formaldehyde required for RIP with CPEB4?

In the past, we compared RIP with and without formaldehyde. There is not a big difference in the result, although with formaldehyde the IP allows for more stringent conditions, which helps when the starting material is limiting and/or prone to RNA degradation. That is why we used it in this case.

16. When using DAVID, what version of the GO database was used?

DAVID 6.8 version was used. It is now specified in the Materials and methods.

17. Why was the February 2014 3'UTR sequences used rather than a more up-to-date version?

The RIP was done several years ago, but more importantly we wanted to compare it to other RIPs that were analyzed using the February 2014 3'UTR sequences. Although it is true that a few specific candidates may be missing because of the old version, the global picture will not change overall. Thus, we considered it better to perform comparable analyses at the expense of having some false negatives. In general, the RIPs are analyzed under stringent conditions in order to avoid false positives at the expense of including false negatives. Thus, the contribution of the dataset version is very minor to the final number of false negatives.

18. SEM are used as error bars when plotting data while SD seems more appropriate.

We have discussed this issue with the Biostatistics Unit of our institute. They argue that both SEM and SD are correct. We have now emphasized the meaning and correct interpretation of the SEM measure, bearing in mind its mathematical relationship with SD, i.e., SEM = SD/√n. We also wanted to clarify that the sample dispersion measure (here SEM) is used in the paper only for visualization purposes.

19. Converting the Socs1 mRNA to show an LPS-response similar to Ptgs2 by mutating CPEs in the Socs1 3'UTR would strengthen the conclusion regarding the hypothesis of the relative number of AREs vs CPEs.

We understand the point raised by the reviewer. If the aim of the work had been to define in detail the behavior of a specific mRNA, we would have performed fine-mapping as suggested. But to define a general rule concerning the relative number of CPEs/AREs, we considered it better to use “synthetic” 3’ UTRs as short as possible and where the elements were in identical backgrounds (Length, AU-content, additional Cis-acting elements, secondary structure, etc.) and inactivated by point mutations (rather than large deletions or inclusions). For example, Ptgs2 3’UTR is 2500 bp long and contains 14 AREs and 8 CPEs. Including these large fragments in Socs1 would also change other relevant parameters, such as 3’ UTR length, as mentioned above, as well as the relative number of CPEs/AREs, and thus make the interpretation more difficult.

Reviewer #2 (Recommendations for the authors):This is a thorough manuscript with an impressive scope of experiments that the authors should be very proud of!

We thank the reviewer for his/her comments and specific points. We have included the requested clarifications as specified in the point-by-point responses.

Specific Comments:(a) Line 95. The authors write: "Cpeb4(deltaM) animals also presented splenomegaly (Figure 1C) and increased splenic levels of the cytokines Il6, Tnf, and Il1a (Figure 1D)." Il6 and Tnf appear to have robust and reproducible effects, but the Il1a effect appears non-significant.

The reviewer is correct, as indicated by the pv < 0.1, it was only a trend (not significant). We thought it may be of interest to the reader to show this trend, it is now shown as ns (not significant).

(b) Line 115. The authors write: "…the levels of anti-inflammatory transcripts, such as Il10 mRNA, were reduced in the Cpeb4 -/- BMDMs (Figure 2H)". Is this really a robust and measurable effect? Error bars on these measurements appear to be overlapping at all timepoints.

This result is the compilation of six experiments (biological replicates). Because the absolute extent of Il10 mRNA induction in each experiment has a high variability, the link between each KO/WT pair is lost and variability is high when compiling the six experiments. However, the difference at the 6 h time point is statistically significant given that in each experiment the difference between the WT and the KO was consistent (we include four representative experiments Author response image 1; Panel A). When we represent individual pairs, at a single time point maintaining the link for each KO/WT pair, the effect of depleting CPEB4 is also clear. Each of these plots loses some information. Originally, we chose the method that contained most information, although visually it is less clear. We would be happy to add or replace the figure by any of the other.

**Author response image 1. sa2fig1:** 

(c) Line 135. The authors write: "We observed HuR binding to Cpeb4 mRNA was strongly enriched after LPS treatment in WT BMDMs but not in p38a(deltaM) BMDMs…(Figures 3D, S3C)." On the contrary, from Figure 3D, it looks to me like there is actually quite strong Cpeb4 enrichment in HuR IPs upon LPS treatment for both WT and p38a(deltaM) BMDMs; untreated WT Cpbe4 looks to be ~0.015 and treated WT ~1.0 (~60-fold enrichment), whereas untreated p38a(deltaM) is very small, well less than 0.005 presumably, and treated p38a(deltaM) is ~0.4 (~80-fold enrichment). Regardless of the exact numbers, there appears to be robust enrichment of Cpeb4 mRNA in both the WT and p38a(deltaM) macrophages upon LPS-treatment.

We thank the reviewer for pointing out this issue. We agree that talking about enrichment with such low levels of binding in untreated samples (levels in the KO are clearly outside the qPCR linear range) is incorrect (ratios between treated and untreated are distorted by this lack of linearity).

We have modified the sentence accordingly:

"We observed that HuR binding to Cpeb4 mRNA was strongly enriched after LPS treatment in WT BMDMs but to a lesser extent in p38a∆M BMDMs, as seen in RNA-immunoprecipitation experiments (Figures 3D, S3D), indicating that this induced binding occurred in a p38a-dependent manner.”

(d) Line 142. The authors write: "…significantly reduced decay rates of both Cpeb4 and Tnf mRNAs in macrophages from mice with myeloid cell-specific TTP deletion" However, from Figure 3F, it looks like there is a large effect for Tnf but actually a much smaller one for Cpeb4. Are both of these decay rates significantly reduced? What are the errors and significance on the RNA decay rates shown in Figure 3F?

Data from Figure 3F was obtained from the available dataset from Sedlyarov and colleagues (https://ttp-atlas.univie.ac.at/wsgi/). From these data, we obtained the n and the FDR, which are now indicated in the figure and figure legend.

(e) I like the luciferase-based reporter assays in Figures 4J and 5K used to demonstrate that the number of CPEs directly influences mRNA stability here; in both cases, definitively linking this to CPEB4 binding (as opposed to other CPEBs or an unknown RBP, for example), would be significantly strengthened by looking for the absence of this affect in a Cpeb4 -/- background. Have the authors considered undertaking these control experiments?

We thank the reviewer for this comment, which we discussed at length in the laboratory when designing the experiment. As shown throughout the study, macrophages develop an aberrant LPS response. Specifically, we show that CPEB4 targets mRNAs are enriched in several proteins of the MAPK pathway, including TTP mRNA (*Zfp36*). Thus, unlike mutations in the cis-acting elements, depletion of CPEB4 has additional effects (such as altered TTP levels) that limit the possibility of drawing clear conclusions about the balance between cis-acting elements. Given that (1) CPEB4 is the only family member that shows alterations in septic patients and in LPS-treated macrophages; (2) (infructuous) previous attempts by our group and others have not identified additional (Non-CPEB) CPE-binding proteins; (3) the CPEB4 KO phenotypes show an aberrant inflammation resolution, we believe it is not far-fetched to attribute the effect of the CPEs to CPEB4.

(f) The last sentence of the manuscript suggests that mRNA stabilization during inflammation results from a dynamic equilibrium between cytoplasmic deadenylation and polyadenylation. This statement is I assume based on the characterized/known function of CPEs and CPEB4 on polyA tails, but there's no data in the current manuscript on polyA tail length changes in response to CPEB4 binding during inflammation in macrophages. Has this been previously characterized in a similar context? Is it a certainty that the CPEB4-dependent effects on mRNA stability measured in this manuscript result from changes in polyA tail length?

This final sentence intended to be an integration of the results of this work with existing literature, where the only described function of CPEB4 is the modulation of the poly(A) tail length. The key point we wanted to raise is that previous studies have considered ARE-mediated deadenylation and destabilization as a binary “end-point” event. In the light of our results, we wish to postulate that the extent of deadenylation (and subsequent destabilization) can be modulated (in extent and time), or even reverted, by the balance between CPEs and AREs. We have clarified this concept.

I think my comments, concerns, and questions listed above are self-explanatory, so I don't have anything additional to add along those lines. This is a dense paper, and I am not a technical expert in every area covered, so forgive me if I missed anything!The one additional comment I have is on whether eLife is the right home for this manuscript:

We believe that our work is of general interest, at the intersection between post-transcriptional regulation of gene expression and inflammatory responses and thus relevant for the broad readership of *eLife*. As an example of this, *eLife* has published over two hundred related articles (including either CPEBs, AREBPs or LPS-macrophages), among these one from our group on the regulation of CPEB4 by phase transitions.